# Applying a systems thinking approach to evaluating the effectiveness of Africa's foodborne disease surveillance systems
Cecilie Thystrup [1] ✉, Tosin Ogunbiyi [2] & Tine Hald[1]

## Abstract

**Background** Foodborne diseases (FBDs) pose a large public health challenge worldwide, particularly in low- and middle-income countries (LMICs), where limited infrastructure, weak regulatory frameworks, and insufficient cross-sector collaboration hinder effective surveillance and prevention. While previous efforts in Africa have focused on risk identification and management training, these approaches often fail to consider the interconnected nature of risk factors, transmission routes, and systemic barriers. As a result, interventions have had limited impact. Addressing this issue requires a systems approach that accounts for the biological, social, and economic complexities of FBD surveillance.

**Methods** We applied the "Five Phase Process of Systems Thinking and Modelling" approach to identify and address the key challenges of implementing effective FBD surveillance systems in Africa. Using leverage point analysis, we identified leverage points with the potential to strengthen the system. We developed and analyzed five scenarios to evaluate the system's performance under various configurations of these leverage points.

**Results** Our approach identifies 33 elements and behaviors that are connected in a systems map through balancing and reinforcing feedback loops. We identify three deep leverage points with the potential to strengthen the system: Public trust, compliance with food safety practices, and data sharing. Among the scenarios assessed, scenario 5, characterized by high levels of public trust, compliance, and data sharing, is determined as the optimal strategy.

**Conclusions** By targeting these elements, countries can pave the way for more effective, sustainable, and culturally appropriate interventions that are critical for improving food safety and overall public health outcomes in these regions. Achieving Scenario 5 will require targeted investments in infrastructure, regulatory support, and public engagement. This study provides actionable insights for policymakers seeking to enhance FBD surveillance systems, contributing to stronger food safety and public health in Africa.

## Plain language summary

Foodborne diseases affect millions of people worldwide with the majority of illnesses and deaths occurring in low- and middle-income countries. To combat foodborne diseases and offer safe food to populations living in these areas, there is a need to approach the problem in ways that look at all the potential elements that play into why foodborne diseases occur and remain difficult to prevent and control. This study focuses on ways to improve food safety monitoring systems in African low- and middle-income countries by identifying the most important factors for success. We focus on three main areas: building public trust, improving hygiene practices among food vendors, and increasing information-sharing between health and food safety sectors. We test different combinations of these factors to see which one would work best in reducing foodborne illnesses. Our results show that a system with high levels of public trust, strict hygiene practices, and strong information-sharing is the most effective. By understanding what makes strong food safety systems, this study provides valuable insights for improving food safety in Africa, which can ultimately lead to healthier communities and reduced risk of foodborne diseases if embraced by policymakers and other stakeholders.

Foodborne diseases (FBDs) have a large impact on public health, especially in low- and middle-income countries (LMIC). The 2015 publication of the first estimates of the Global Burden of Foodborne Disease Report[1] emphasized the significance of FBDs as a 'silent pandemic', necessitating a concentrated effort towards mitigation of the associated health burden[2]. FBDs are responsible for acute illnesses and long-term health complications, which often place a disproportionate burden on vulnerable populations in LMICs[3,4].

Effective surveillance systems for FBDs are crucial for early detection, response, and control of outbreaks and mitigation of risk[2,5]. These systems can cover a variety of elements, such as routine laboratory testing, monitoring of specific pathogens, and the integration of advanced molecular

[1]National Food Institute, Technical University of Denmark, Kgs, Lyngby, Denmark. [2]Department of Biological Sciences, Mountain Top University, Ibafo, Ogun State, Nigeria. ✉e-mail: ceth@dtu.dk

technologies such as Whole Genome Sequencing (WGS) and metagenomics[6,7]. For instance, many countries in the European Union (EU) have implemented robust surveillance systems for foodborne pathogens such as *Campylobacter* spp., and non-typhoidal *Salmonella*, which include systematic sampling and sharing of real-time data across multiple countries[8,9]. Because of how these systems are designed, different organizations are able to assess trends and detect outbreaks and facilitate timely public-health responses, which helps to reduce the incidence and spread of FBDs[8]. Such surveillance systems require substantial investment in infrastructure, trained personnel, and inter-organizational coordination, as well as consistent political and financial support to maintain their effectiveness[10,11].

Initiatives like the Integrated Disease Surveillance and Response (IDSR) framework for Africa have also been prepared to strengthen national capacity for public-health surveillance and outbreak response, but despite the availability of the IDSR guidelines, many countries continue to face challenges in public-health surveillance systems[10]. Inadequate or ineffective reporting systems, inadequate regulation of consumed food, particularly in the informal sector, and large-scale, consumption of contaminated foods have been linked either directly or indirectly to the persistently high incidence of FBD illnesses in many LMICs[12–15]. In this context, the informal sector refers to segments of food production and distribution that operate outside formal regulatory and commercial frameworks, including food that is locally produced and sold at open markets or consumed within households, without entering industrialized food supply chains. In addition, limited resources and infrastructure further complicate efforts to monitor, control, and prevent FBDs effectively[12–15].

Other challenges include poor data-sharing protocols, which directly limit the implementation of coordinated outbreak responses[11,16]. In many African countries, data sharing between governmental sectors is limited. This makes effective FBD surveillance and response difficult, as it typically requires concerted actions from multiple responsible sectors[11]. The informal nature of how many food products are produced or sold in Africa also poses a very large challenge in monitoring FBDs. This is because of the minimal regulation, low compliance with food-safety practices, and limited traceability, making it difficult to trace sources of FBDs and implement mitigation efforts to prevent future incidences[12,17]. The threat of antimicrobial resistance (AMR) in developing countries complicates the challenge of FBD surveillance even further by making it harder to identify and correctly treat infections caused by AMR pathogens. Detecting these infections often requires advanced diagnostics that are limited in LMICs[18].

Research on FBDs and the introduction of various programs aimed at studying the epidemiology, control, and prevention of FBDs in Africa have raised awareness among African governmental agencies[19]. Past efforts have included management training and identification of potential risk factors or transmission routes, which are important elements, but these approaches often fail to understand how these elements play into a larger system and are affected by systemic barriers[20].

The problem of FBD surveillance can be described as a "wicked problem," where the effects of the problem are interlinked in ways that involve contradictory and changing requirements[21–23]. This means that there is no determinable place in the problem where a single solution can be made, thereby increasing the number of potential solutions[23]. Such 'wicked problems' can have large, long-term effects on social, economic, or environmental elements[23]. They can also include time lags, where there will be a delay from the onset of the cause to the effect[23]. In many African countries, the link between the contamination of food products and the observed increase in illnesses is rarely established due to the absence of effective surveillance systems[24,25]. While the time lag between contamination and health outcomes may be shorter than in other complex health challenges, the inability to detect, attribute, and monitor such events undermines both the recognition of the problem's scale and the ability to evaluate the effect of any intervention efforts[26]. As a result, even effective prevention measures may go unnoticed, reducing the perceived incentive to invest in FBD control.

Systems thinking approaches can help to address FBD surveillance in Africa by improving the understanding of how different components within the FBD surveillance system interact dynamically[27]. By taking a systems thinking approach, the complexity of underlying behaviors is broken down to independent elements that can be assessed, thereby enabling the identification of leverage points important for the function—or lack of function—of the system[28,29]. Its significance in offering new opportunities for comprehending and continuously refining our understanding of the dynamic nature of each system's components, including how to intervene effectively. Using it to assess the problem of establishing effective FBD surveillance in Africa could enrich our understanding of the complexities and show which targeted changes would have the greatest impact on the system[27,30]. Determining system dynamics, such as causal loops, provides insights into the relationships among system components. Causal loops are structures that cause system dynamics, revealing patterns of change over time, revealing which elements merit continuation and those that necessitate reevaluation[28,31]. By embracing a systemic perspective, policymakers can develop more effective, sustainable intervention strategies to mitigate the burden of FBDs in African countries[32,33].

Systems thinking has been evident in the understanding of other "wicked problems" such as AMR and outbreak response[30,34]. It has, to the best of our knowledge, not been used as a mapping tool to identify interventions that can initiate a system-wide change for FBD surveillance in Africa. This study aims to identify strategic intervention points to enhance the effectiveness and sustainability of FBD surveillance systems in Africa by addressing the following research question: How do structural, behavioral, and contextual barriers limit the implementation of effective FBD surveillance, and what opportunities exist to strengthen these systems? Our approach identifies 33 elements and behaviors that are connected in a systems map through balancing and reinforcing feedback loops. Three deep leverage points have the potential to strengthen the system: Public trust, compliance with food safety practices, and data sharing.

## Methods
### Overall approach
In this study, we applied the "Five Phase Process of Systems Thinking and Modelling" approach from Maani and Cavana[28] to systematically explore and address the challenges hindering the implementation and effectiveness of FBD surveillance systems in Africa (Fig. 1). To capture a comprehensive view of the issue, we conducted a collaborative workshop with a diverse group of stakeholders, including researchers, health workers, legislators, and other professionals actively involved in FBD surveillance in different African countries. During this workshop, participants used the Iceberg model,[28,35] a framework that highlights how much of a problem remains hidden beneath the surface, to assess FBD surveillance from their respective contexts, responding to each of the four questions to identify key elements affecting FBD surveillance effectiveness[28,35]. The workshop responses were organized into a structured list of influential behaviors and factors, covering three main domains: societal dynamics, the healthcare system, and food and agricultural practices.

Following the workshop, we conducted a validation step to ensure the reliability and coherence of the identified elements. We performed a coherence analysis that served as an internal validation process, where we compared the key elements from the Iceberg Model across the different participant groups in the workshop. This was done to ensure that the connections within the model were clear, logically consistent, and free of contradictions. We then conducted external validation by triangulating our methods: in addition to the workshop findings, we conducted a literature search on foodborne disease surveillance systems in Africa, comparing the results with the workshop insights to confirm alignment and increase credibility.

The elements identified in the workshop served as a reference for the next phase, in which we developed causal loop diagrams (CLDs). These diagrams visually represented the interconnectedness of the identified elements, allowing us to map out reinforcing and balancing feedback loops that contribute to the persistence of FBD challenges in Africa. Next, we performed a validation step,

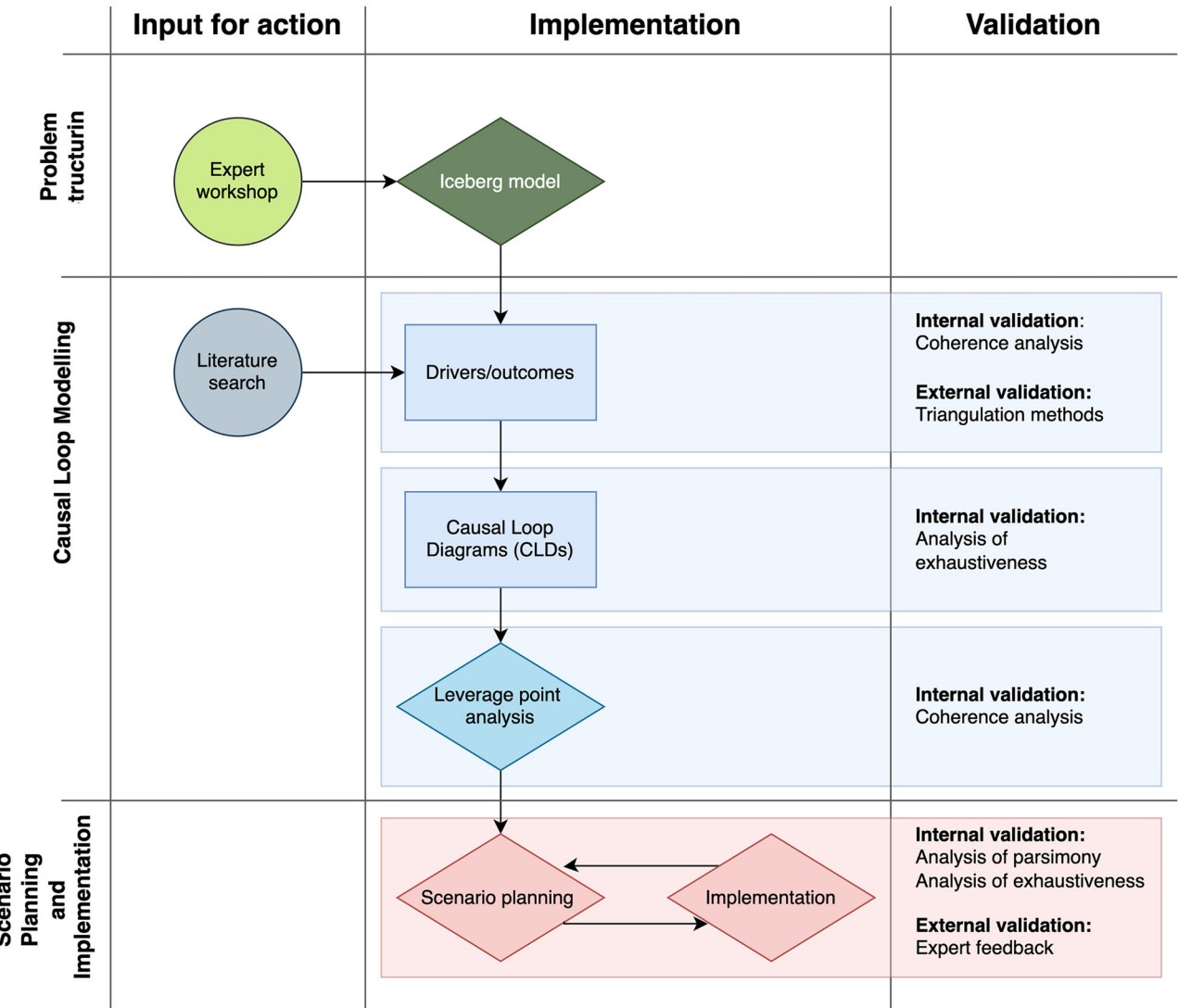

**Fig. 1 | Qualitative flowchart diagram of the systems thinking approach following the methodology from Maani and Cavana[28].** Color shading is used to distinguish the different phases of the approach: green tones represent the problem structuring phase, blue tones correspond to the causal loop modeling phase, and pink tones indicate scenario planning and implementation.

in which each feedback loop and variable was systematically reviewed to confirm that all relevant variables, interactions, and feedback loops were represented. This process involved cross-checking each loop and variable against known factors in the FBD surveillance system.

After the validation step, we conducted an "archetype" identification process to further analyze the CLDs. Following the approach described by Senge[35], we systematically reviewed the structure and interactions of the feedback loops to detect recognizable system archetypes that represent common patterns of system behavior. Identifying these archetypes helped clarify how recurring challenges and systemic bottlenecks arise within the FBD surveillance system. Furthermore, this analysis was used to highlight structural limitations that perpetuate ineffectiveness in the system and to identify strategic points where targeted interventions could disrupt problematic cycles and improve overall system effectiveness.

With a better understanding of the system structure as well as the dynamics from the archetypes, we performed a leverage point analysis. Specifically, we examined which variables were central to the reinforcing or balancing feedback loops and assessed which interventions could potentially disrupt the negative patterns detected in the archetype analysis.

The final steps of the analysis were the scenario planning and implementation steps, based on the methodology described by Maani and Cavana[28]. Building on the results of the leverage point and archetype

analyses, we developed a set of scenarios, which included defining the general scope, time frame, and boundaries of the scenarios. We then identified key drivers of change, critical uncertainties, and external factors that could impact the FBD surveillance system. Using these drivers, we constructed a range of plausible future scenarios and evaluated the performance of potential strategies across different scenarios, both positive and negative. Strategies were assessed against a set of performance criteria to determine their robustness and ability to meet surveillance system objectives. Following the scenario planning, a validation step was implemented. We conducted an analysis of parsimony, where we reviewed the models and scenarios to ensure they contained only essential variables and feedback, and an analysis of exhaustiveness, where we verified that all critical elements and interconnections relevant to the system were adequately captured. We use the terms analysis of parsimony and analysis of exhaustiveness to describe the methodology of minimizing unnecessary complexity and ensuring completeness of system representation, which are grounded in the principles outlined in the systems thinking approach[28,36]. These internal validation steps were supplemented by expert feedback: results from the leverage point analysis and scenario planning were reviewed by the workshop experts to assess the feasibility, relevance, and completeness of the proposed strategies. This validation helped ensure that the final recommendations were grounded in both empirical system behavior and expert judgment.

## Applying the Iceberg model in a workshop

To explore the challenges of establishing FBD surveillance in Africa, we organized a problem-structuring workshop in Arusha, United Republic of Tanzania, in February 2024. This workshop was held as part of a meeting conducted by the "Foodborne Disease Epidemiology, Surveillance and Control in African LMIC" (FOCAL) group, which involved a diverse group of participants directly involved in FBD surveillance in Africa countries (Supplementary Data 1). These participants were chosen because of their diverse backgrounds and first-hand experience with foodborne diseases in Africa, and their backgrounds included healthcare professionals, researchers, representatives from governmental agencies, and stakeholders from Nigeria, Tanzania, Mozambique, Ethiopia, and South Africa. The participants represented a range of expertise in FBD surveillance spanning fields like epidemiology, healthcare management, food safety, and public health policy. Participants varied in age, gender, and professional background, contributing to a comprehensive perspective on FBD surveillance challenges across the region.

The workshop was designed to leverage participants' practical knowledge and expertise in surveillance. In initial sessions, participants were divided into groups to discuss FBD surveillance challenges using the Iceberg model framework, a systems thinking tool commonly applied to understand how visible events are shaped by deeper patterns, structures, and underlying mental models[28,35]. Each group worked through a series of guiding questions (Supplementary Table 1) to identify key stakeholders and major challenges. They also served to guide the participants into classifying the elements identified into societal, healthcare and food, and agricultural categories, based on their impact. The insights gathered were systematically analyzed by organizing responses into themes and mapping them within the Iceberg model, organizing observations into four layers: observable events, patterns/trends, underlying structures (e.g., infrastructure, policies), and mental models (i.e., deeply held beliefs that influence system behavior). Preliminary findings were shared back with participants for validation and refinement, ensuring that the analysis accurately reflected their perspectives and expertise.

This structured approach enabled participants to formulate context-specific issues and examine them within broader dynamics. Workshop proceedings were documented by recording each group's responses to the Iceberg model questions, with specific attention given to the different layers identified by each participant group. Summary notes were taken for each group's iceberg model, capturing the key elements, patterns, structures, and mental models they identified.

To address potential bias, we ensured that discussions were facilitated in a way that encouraged open contributions from all participants, reducing the influence of any single perspective on the collective output. In addition, each group's data was independently reviewed by the facilitators to identify any outlying perspectives and ensure consistency across responses. This approach allowed us to capture a balanced view of the system's dynamics without any single participant's background influencing the results.

## Scoping review

To validate and qualify the insights, a scoping review was conducted to compare the findings of the workshop with existing research on FBD challenges. Based on this, a series of search strings was developed and used to systematically search PubMed for relevant literature. Each search was conducted using PubMed and tailored to reflect a specific concept related to the elements identified in the participant workshop. If a search returned more than 20 results, only the first 20 hits were screened by title and abstract for relevance. Literature was considered relevant if it described structural, behavioral, or contextual challenges related to the elements identified by the participants in the workshop. In this study, structural challenges refer to the barriers in the formal systems, such as governance structures, legislation, and resource distribution. Behavioral challenges involve the actions, choices, and incentives of the individuals or groups that affect the system, and contextual challenges include broader environmental or societal factors like culture, normal, or political context that shape how systems operate. A

summary of the search terms, number of hits, and included publications is provided in Supplementary Data 2.

## Defining system elements and boundaries with CLDs

To assess the challenges hindering the implementation and effectiveness of FBD surveillance systems in Africa, we developed causal loop diagrams (CLDs), based on the elements identified as part of the Iceberg model. CLDs are visual tools used to illustrate complex issues through the use of feedback loops between different components within a system, which can either reinforce (positive loops) or balance (negative loops) system behavior[28,29,37]. The system boundaries were fixed so that only elements that were a key part of the system were included to avoid extending the system indefinitely[38]. The purpose of the system was identified in the context of FBD diseases and was thus defined as the "challenges hindering the implementation and effectiveness of FBD surveillance in Africa". Feedback loops were identified using the terminology from "Thinking in Systems: A Primer"[29], which denoted the elements as inside, outside, or external to the system, and the interconnections as either reinforcing (positive) or balancing (negative) feedback loops. To gain further insight into recurring patterns within the system, we conducted an 'archetype' analysis. System archetypes, commonly encountered structures that reveal persistent problems or unintended consequences, were identified using the approach described in "System Archetype Basics: From Story to Structure"[36].

## Leverage point perspective and system dynamics

Following the development of the CLDs and archetype identification, we conducted a leverage point analysis to identify key areas within the FBD surveillance system where targeted interventions could produce the greatest impact. Leverage points are locations within a system where small, strategic changes can lead to substantial system-wide effects[31].

We then evaluated potential intervention points by analyzing variables central to the major reinforcing and balancing feedback loops identified in the CLDs. Using the classification system proposed by Meadows, leverage points were categorized along a hierarchy, ranging from shallow leverage points that can be used to adjust the operational aspects of a system (layer 4–12), to the deeper leverage points that can fundamentally transform the system (layer 1–3)[31]. Different interventions that intend to improve FBD surveillance in Africa and mitigate disease were listed and categorized, based on the level of impact of the leverage points[39,40].

## Scenario planning and implementation

We applied the methodology from Maani and Cavana[28] to evaluate the policy options for strengthening the FBD surveillance system. Following the identification of deep leverage points from the dynamic modelling and leverage point analysis, we used scenario planning to explore how these interventions would perform under various conditions[28].

First, we identified key drivers of change relevant to the FBD surveillance system, such as changes in public health infrastructure, food safety practices, political commitment, and inter-sectoral data sharing mechanisms. For each scenario, we established the scope, timeframe, and boundaries: The scope focused on public health interventions, food safety practices, and inter-sectoral data sharing; the timeframe covered a 10-year timeline to capture both short-term impacts and long-term system adaptations; and boundaries were set to include only policy-relevant factors directly influencing FBD incidence, surveillance capabilities, and public cooperation.

Each policy strategy was then assessed across different performance measures such as resilience, adaptability, and overall effectiveness in reducing FBD incidence. This was done in order to select the strategies that showed the highest potential for long-term improvement across the key drivers of change. To select the optimal scenario, we followed a structured validation process to ensure the model's relevance and robustness. We conducted a parsimony analysis, which involved simplifying the model to only contain the deep leverage points identified in the leverage point analysis, and an analysis of exhaustiveness, where we assured that all critical

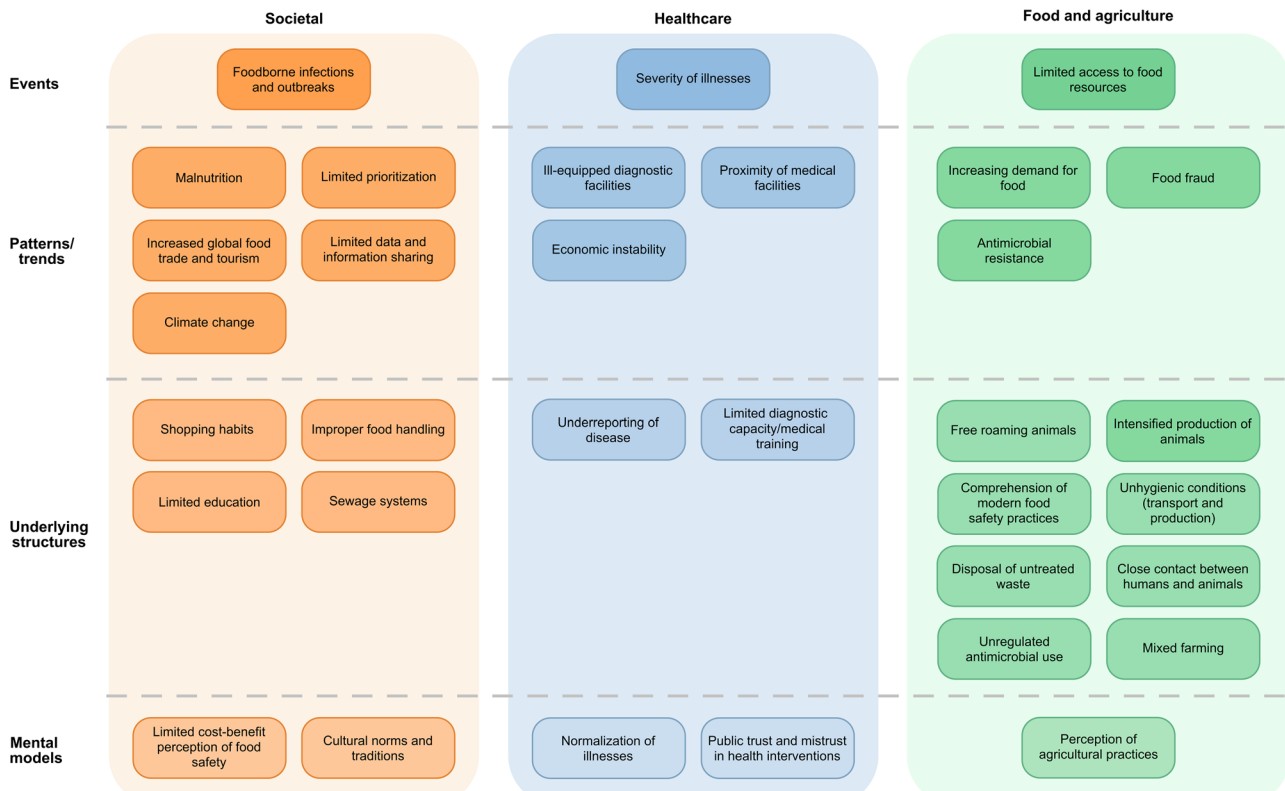

**Fig. 2 | Elements and behaviors identified in the expert workshop organized by the Iceberg model.** Additionally, color shading is used to indicate the primary domain in which each element has the greatest impact: broader society (orange), the healthcare system (blue), and the food and agricultural sector (green). The literature search conducted as part of the validation of the results is referenced in Supplementary Data 2.

elements were included. This was carried out by identification of key elements and drivers in the current scenarios, which could influence the model. Finally, we validated the scenarios externally by presenting the scenarios to the same group of experts in FBD surveillance and public health from the workshop afterwards.

### Ethics declarations

Our research prioritizes ethical considerations and inclusivity by involving experts and stakeholders from relevant African low- and middle-income countries (LMICs). These experts, all authorities in their respective fields, contributed valuable insights and have been acknowledged in Supplementary Data 1. All participants provided verbal informed consent to participate in the workshop. This collaborative approach ensured that the research was both comprehensive and culturally relevant, reflecting the diverse perspectives and expertise of those directly impacted by the study. Ethical approval was not required for this study because it presents a conceptual framework based on findings from a stakeholder workshop conducted with participants already engaged in the project. No personal or sensitive data was collected. All other data used in the study were derived from publicly available literature and a scoping review of published articles.

### Reporting summary

Further information on research design is available in the Nature Portfolio Reporting Summary linked to this article.

### Results

#### Problem structuring using the Iceberg model

The foodborne disease workshop for stakeholders identified 33 elements and underlying behaviors influencing the current system of FBD-surveillance in Africa. The Iceberg model was employed to evaluate the interplay of these elements within the existing system of FBD surveillance in Africa. At each level, these factors were

categorized based on their primary influence in society, the healthcare system, or the food and agricultural sector (Fig. 2). "Events" (i.e., outcomes that are visible and above the waterline in the iceberg) included those related to foodborne disease infections, severity of illnesses, including hospital admissions, and access to food resources. These outcomes were consistently reflected in both the stakeholder workshop and the scoping review findings (Supplementary Data 2). Additional literature identified outside the scoping review further supported the relevance of these observed outcomes[2,5,17,41,42]. "Patterns/ trends" included the increasing demand for food and the inadequate prioritization of surveillance for locally consumed food sold in the informal sector, ill-equipped diagnostic facilities, proximity of medical facilities, limited data sharing and reporting, climate change, and food fraud. These patterns were frequently cited by workshop participants and were further substantiated through the scoping review and complementary literature sources[13,43–60]. The "Underlying structures" were identified as limited education, human- and animal interactions (i.e., free-roaming animals, proximity of livestock and crops (mixed farming), close contact between farmers, their families, and animals). Other foundational challenges identified were antimicrobial resistance and usage, cost of treatment, access to sanitation, unhygienic conditions in slaughterhouses and transportation of farm produce, and disposal of untreated agricultural waste. These structural drivers were consistently reflected in both the participant discussions and the broader body of literature consulted[2,5,12,13,17,21,54–56,61–82]. Finally, the 'Mental models' were described as primarily driven by stigma, political- or societal knowledge, and inadequate prioritization and preservation of age-long agricultural practices, e.g., free-roaming animals, which was supported by the scoping review and the complementary literature sources[5,21,67,83–85]. Participants in the workshop described the increasing population as well as limited information as central focus for interventions. They emphasized that, as the increasing population

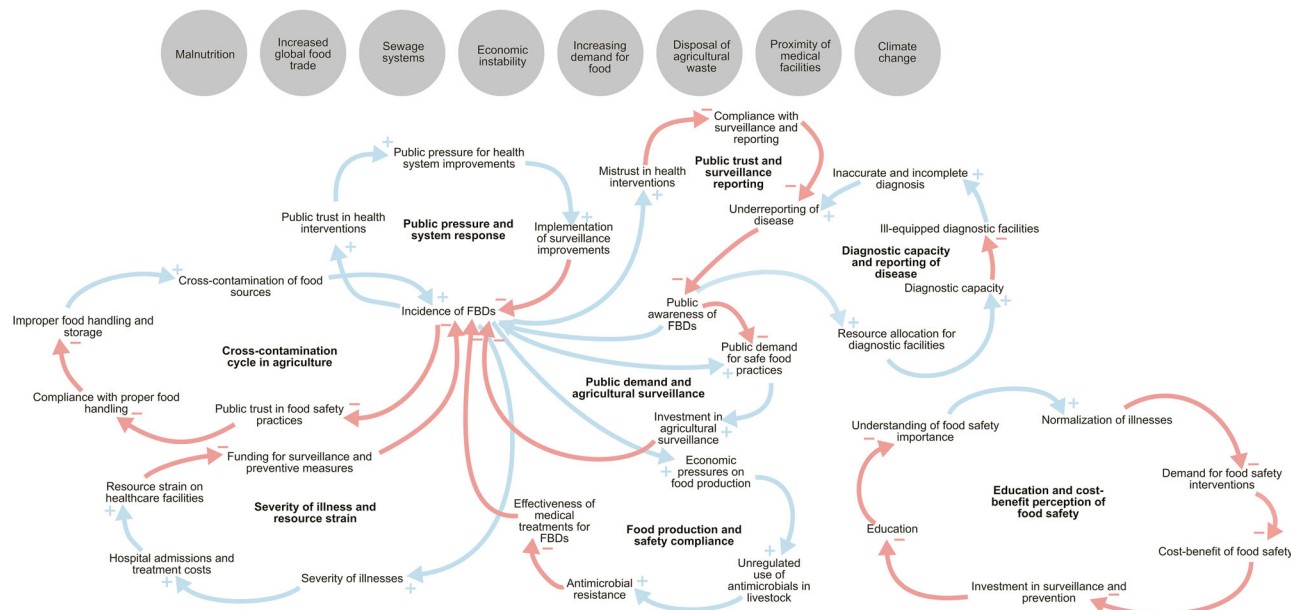

**Fig. 3 | Current system of elements connected to foodborne infections and disease outbreaks in Africa.** Gray circles depicts elements outside of the system. Arrows indicate directional relationships between elements: red arrows signify negative or decreasing effects, while blue arrows indicate positive or increasing effects.

intensifies the demand on resources, there is a need for sustainable solutions that will be able to work long-term.

### Scoping review results

The purpose of this review was to assess whether the challenges raised by participants were reflected in existing research and to identify any additional elements or overlooked issues that were not mentioned during the workshop. The search strings, number of hits, and corresponding curated publications are summarized in Supplementary Data 2. Publications deemed relevant were recorded and are presented in the Supplementary References.

### Causal loop diagrams and archetype definitions

The elements identified in the Iceberg model were classified as internal, external, or outside elements and were displayed in an interconnected map (Fig. 3). To limit the system to elements that had a direct impact on the present FBD surveillance system, outside elements were simplified when necessary. This was the case for the elements "Climate change," "Malnutrition," "Sewage systems," "Economic instability," "Increasing demand for food,", "Disposal of agricultural waste," and "Proximity of medical facilities," which were identified as important drivers outside the system.

The CLDs were structured by the primary categories of influence on FBD surveillance: societal factors, healthcare system factors, and food and agriculture factors. Each CLD represents a set of interconnected feedback loops within these categories, revealing how specific variables reinforce or counterbalance challenges within the surveillance system. For example, reinforcing loops may perpetuate existing issues by amplifying factors like underreporting and limited diagnostic capacity, while balancing loops can contribute to system stability by counteracting such changes through public pressure or policy responses.

The key variables included "Underreporting of disease," "Limited education and awareness," "Resource allocation," "Mistrust in public health and regulatory systems," and "Antimicrobial resistance". These were chosen because they represent critical points of influence in the system across all three categories, which either directly or indirectly impacted the shape of the system's response to FBD and their ability to detect them. System dynamics show that these elements are connected to each other through several reinforcing or balancing feedback loops. The CLDs for each individual loop are available in the Supplementary Information (Supplementary Figs. 1–8).

The loops named "Public trust and compliance with surveillance" and "Public pressure and system response" are both balancing loops that illustrates how the public perception can both hinder and harbor the implementation of FBD surveillance systems; rising FBDs can lead to mistrust in health systems, reduced compliance and surveillance reporting, which then exacerbates the underreporting of disease and hides the true extent of FBDs, thus perpetuating the issue. On the other hand, increased incidence of FBDs can also drive public pressure on health authorities to improve surveillance, which can lead to reduced FBD incidence and restored trust in the future. Similarly, the loop named "Public demand and agricultural surveillance" shows how underreporting initially limits awareness and investment in agricultural surveillance but can eventually drive public demand for safer practices and surveillance improvements, as the FBD rates increase.

The loop named "Diagnostic capacity and disease reporting" shows that underfunded diagnostic facilities contribute to underreporting, which leads to decreased awareness of FBDs and consequently less resource allocation, further worsening diagnostic capacity.

These loops are all balancing loops that keep the system in place by counteracting change and resist drastic changes.

Reinforcing loops included "Education and cost-benefit perception of food safety" and "Cross-contamination cycle in agriculture", which showed how the limited education reduces the understanding of food safety, which normalizes FBD illnesses and lowers the public demand for food safety measures, thereby reinforcing low investment in FBD surveillance. Similarly, improper handling of food leads to contamination, which increases FBDs and decrease public trust, which in turn reduces compliance and further perpetuating unsafe food practices.

Other reinforcing loops included "Severity of illness and resource strain," where the severity of illnesses increase the hospital admissions and puts increased strain on resources, which diverts funding away from surveillance and prevention, and leads to increased FBD incidence, and the "Food production and safety compliance" loop, which show how economic pressure can encourage antimicrobial use, leading to resistance and worsening of FBD diseases, and further economic pressure on food production systems. These loops lead to gradual worsening of FBD incidence or severity because of resource strain, low compliance, and economic pressures compound issues.

Archetypes of the system were identified within each loop to help clarify how certain behaviors were perpetuated and highlight how common

structural challenges could hinder the effectiveness of a FBD surveillance system. Several recurrent patterns emerged within the FBD surveillance system (Supplementary Data 3). The "Fixes that fail" archetype highlighted how short-term responses, such as temporary increases in diagnostic resources during outbreaks, fail to create sustainable diagnostic capacity improvements, leading to repeated gaps in disease detection. This was also the case for the "Escalation" archetype, where the increased severity of FBD leads to healthcare resources becoming more strained and creates a cycle of escalating demand on an already strained system. Similarly, the "Drifting goals" archetype showed how inconsistent resource allocation and low public trust have contributed to reduced compliance with food safety standards, normalizing high levels of foodborne disease incidence.

The "Limits to success" archetype showed how growth in diagnostic capacity may initially improve, since resources are allocated to the problem, but as the system evolves, financial constraints, limited skilled personnel, or outdated infrastructure begin to slow the progress further. This prevents the diagnostic capacity from expanding, thus hindering the system's ability to effectively monitor and respond to FBDs. This archetype goes together with the "Growth and underinvestment" archetype, which describes how a system's growth is limited due to insufficient investment in critical infrastructure. If FBD risks are underreported, the public's awareness associated with food is reduced, which removes the pressure to prioritize food safety measures in agriculture. Without the pressure for legislators to invest in FBD surveillance, the system continues to miss effectively addressing the problem, which leads to delayed responses and limited surveillance capabilities. Same with the "Shifting the burden" archetype, where the resources are often allocated to immediate, symptomatic responses to outbreaks rather than a comprehensive education in food safety, thus reinforcing the perception that food safety practices are costly and not worth prioritizing.

"Tragedy of the Commons" showed how the inadequate regulation of informal food vendors can lead to increased incidence of FBDs when vendors decide to bypass safety measures, ultimately harming both consumers and the sellers.

These archetypes emphasize the underlying issues that prevent the effectiveness of surveillance efforts, suggesting areas where targeted, long-term interventions could disrupt these patterns.

## Leverage points of the FBD systems map

Building on these archetypes, we identified several leverage points where targeted interventions could effectively disrupt the reinforcing loops that hinder the implementation of effective FBD surveillance systems in Africa. Deep leverage points included "Public trust in health interventions," "Compliance with food safety practices," and "Data sharing across sectors". These points were classified as deep leverage points because they were determined to create important structural and behavioral changes in the FBD surveillance system.

Public trust is central to the feedback loops, and they influence compliance, reporting, and overall public cooperation, so changing the level of public trust would impact the public's willingness to participate in surveillance initiatives and follow food safety guidelines. Similarly, compliance affects the spread of pathogens and the overall safety of food production, which means that targeting this point will affect the core behaviors around food handling and safety practices.

Improved data sharing will support coordination among healthcare workers, the public sector, and the agricultural sector, thereby enhancing a system-wide response. This point restructures the information flow across sectors, enabling collaboration between different sectors. These points affect the core paradigm around health interventions and public engagements.

These components suggest that improving the public understanding of safe food practices, acknowledging cultural norms and knowledge on sustainable agricultural practices, is crucial to ensure a system-wide change. In many African communities, traditional beliefs and practices often influence health behaviors, like reliance on traditional healers, who are sometimes trusted over modern medical practices[67,68]. This belief can lead to seeking remedies from traditional healers instead of professional healthcare

providers, which complicates efforts to report, manage, and control FBD infections effectively. Additionally, there may be resistance to visiting hospitals due to mistrust in modern medicine, which is often influenced by past negative experiences, or the stigma associated with certain illnesses[67]. This emphasizes how the "limited education" element indicates a profound need for paradigmatic shift in how food safety knowledge is utilized and communicated. If the mindset is shifted towards awareness and prevention, it could lead to more effective implementation of food safety practices. This is essential for long-term sustainability and effectiveness, because they address the root causes of the FBD surveillance system and point to public health vulnerabilities.

Shallow leverage points included "Diagnostic capacity", "Resource allocation for surveillance", "Antimicrobial resistance control", "Severity of foodborne disease", and "FBD incidence". These points can all help to provide important improvements within the existing system, but do not fundamentally alter the system's goals, structures, or paradigms. They point to regulatory changes that need to happen to enhance facility equipment and optimize public health logistics, which are crucial for improving accessibility to medical services. While allocating funding is a straightforward starting point, it is essential to integrate other elements within the communities to ensure sustained efforts. Engaging local communities in recognizing FBDs can facilitate the establishment of surveillance strategies that capture general trends across diverse communities and help prioritize resources to areas most in need.

## Scenario planning and implementation

Building on the leverage point analysis, we approached the scenario planning process through a series of steps, including the identification of key drivers of change, definition of scope, timeframe, and boundaries, as well as assessment of each scenario against performance measures. To clarify, key drivers were identified based on insights from the CLDs and the leverage point analyses. The following drivers were selected based on their influence on the FBD incidence, effectiveness of the system, and public engagement:

Public trust influences the public's willingness to comply with food safety guidelines, report cases of FBDs, and engage with public health interventions. Trust levels fluctuate based on the transparency in health communication, visibility of legislation efforts such as rules or guidelines, and public perception of health authorities' effectiveness. Scenarios were designed to test variations in the public trust to understand how changes in this driver could affect compliance and data sharing.

Compliance with food safety practices is directly related to the effectiveness of a successful FBD surveillance system, which is why compliance was identified as a key driver on the risk of contamination and, subsequently, the incidence of FBDs. Scenarios were designed to test the effects of high, medium, and low compliance across various settings, particularly in informal settings where legislation is often limited.

Effective data and information sharing practices between different sectors, such as public health, agriculture, and food, are very important for coordinating outbreak detection and response. It was identified as a driver since data sharing enables faster response times and collaboration. Scenarios were designed to explore different levels of investment in digital infrastructure and reporting to assess how improvements in data sharing could enhance system resilience.

Each scenario was constructed to simulate varying conditions based on the identified key drivers of change, within the scope, timeframe and boundaries defined as scope, timeframe and boundaries.

The scope focused on the three components identified in the leverage analysis, due to their direct impact on FBD incidence and system responsiveness. The scenarios simulated how shifts in each of these areas would influence the effectiveness of FBD surveillance.

The timeline was determined to be 10 years in order to capture both immediate and long-term effects. This timeframe should allow for observation of short-term responses like changes in FBD incidence following, e.g., a change in legislation, and long-term effects, like sustained improvements due to improved compliance and data sharing.

The boundaries were set to focus on policy-relevant factors affecting FBD surveillance, excluding anything beyond the control of the system (e.g., global warming and economic crisis). These boundaries ensured that the scenarios remained grounded in actionable policies within the FBD surveillance system.

Each scenario was then evaluated against effectiveness, adaptability, and resilience. An overview of the different scenarios and their impact on the three key drivers of change, as well as their effectiveness, adaptability, and resilience, is available in Table 1.

Through the evaluation of each scenario, scenario 5 was determined to be the most robust option for enhancing FBD surveillance in Africa. Scenario 5 combines high levels of public trust, compliance with food safety practices and data sharing across sectors, achieving optimal results across both the effectiveness, adaptability and resilience criteria. Although this scenario represents ideal conditions, it also presents some challenges, especially in African contexts where limited IT infrastructure and resources may restrict capabilities. Addressing these constraints through targeted investments in infrastructure and capacity building would further strengthen the implementation of scenario 5, making it feasible for a long-term impact.

## Discussion

The combined knowledge from the workshop and the current scientific knowledge shows that the present system requires deep changes to mitigate further risks of FBDs. Frequent occurrences of "Foodborne infections" can lead to an increased "Normalization of illness." This normalization reduces the willingness to allocate resources and implement efforts to combat these diseases, potentially resulting in a higher incidence of foodborne diseases, thereby reinforcing the feedback loop. By enhancing knowledge and changing behaviors within the general community at the consumer level, exposure to foodborne pathogens could be reduced, as educated food producers and handlers, as well as consumers, are more likely to practice safe food handling, leading to decreased disease transmission[20]. Similarly, consistent engagement of local communities in disease surveillance and prevention efforts could empower them to actively participate in the monitoring, reporting, and prevention of disease. This can create a positive feedback loop, which in turn can drive community-based initiatives that enhance disease awareness and encourage collective action. This approach has already demonstrated effectiveness in a study from Mozambique, where local stakeholders collaborated to develop a strategy for FBD prevention[86].

The intensification of livestock production driven by increased food demand and profit may result in increased use of antibiotics, subsequently leading to antimicrobial resistance (AMR). AMR makes infections more difficult to treat, potentially increasing the severity of illnesses as more infections become resistant to antimicrobial treatments[87–89]. In traditional farming, the proximity of livestock and crops can lead to increased spread of pathogens from animals to crops or plants ("Mixed farming" and "Free-roaming animals"), which can lead to more infections[63,72]. Some animal farming practices establish a close contact between farmers, their families, and their animals, which increases the risk of disease transmission between humans and different food animals and food products, which consequently could lead to more contaminated food on the market ("Cross-contamination of food")[2,5]. Implementation of hygiene protocols and related training will improve educational knowledge, which will make the farmers and food producers more likely to adopt practices that reduce the spread of diseases[21].

Insufficient reporting and data sharing can exacerbate food fraud because fraudulent practices often go unreported or concealed, undermining the integrity of food safety data[13]. Poor data on food safety can lead to under-prioritization of the area compared to other public health issues. As food safety becomes less of a priority, the risk and occurrence of foodborne infections may increase, feeding back into the cycle and highlighting the systemic weaknesses caused by initial food fraud. Educating consumers about the signs of food fraud could lead to increased ability to detect and report fraudulent practices, which can help maintain the integrity of food safety data. Similarly, if this data is shared between consumers, regulators,

and public health professionals, the information can be used to improve detection and response to food fraud[13,52].

Inadequate prioritization of FBD surveillance by governments can lead to lenient oversight of food safety, influencing "shopping habits" where consumers may unknowingly purchase foods that are at risk of being contaminated. This behavior increases the likelihood of "improper food handling, including cross-contamination at markets", which in turn contributes to the number of foodborne illnesses in the population. As the extent of food contamination (e.g., prevalence) becomes visible through reporting, this can prompt a reevaluation and increased prioritization of FBD surveillance, thereby potentially reducing the initial complacency and closing the loop. Informed and educated consumers can make safer choices, reducing the risk of purchasing contaminated food. Safer market environments will also decrease the likelihood of disease transmission by enforcing strict hygiene standards at food vendors and markets. A good example of a system like this is the rating system used by the Danish Veterinary and Food Administration (Fødevarestyrelsen) to evaluate and display the hygiene standards of food establishments, such as restaurants, cafes, and supermarkets. The system helps ensure food safety and informs consumers about the hygiene standards of the places they choose to eat or buy food from, using a smiley that is displayed and visible to the public[90]. Implementing a rating system like this would be a powerful tool for reducing FBD transmission by setting clear expectations for hygiene standards and help sustain a culture of food safety across markets and vendors.

In the current system, increased foodborne disease infections raise the importance of food safety, leading to more resources being allocated to food safety. In turn, this reduces the incidence of infections. On the other hand, if food safety is deprioritized, it leads to more infections, which again could shift focus back to food safety. This dynamic seeks to balance the level of attention and resources dedicated to food safety based on the current state of public health concerning FBD. Enhancing public awareness about food safety and the risks involved in improper food handling could make consumers more likely to demand safe food practices, which can influence policymakers and allocate resources accordingly. Similarly, robust policies ensure consistent attention to food safety, which prevents complacency when FBD infections are not so visible in the population. If food producers or market vendors were held accountable for safety standards, the burden of infections on public system might be reduced[25].

Therefore, there is a need to change the system, where the appropriate leverage point to enact change to the structure of the system is in transforming the mindset or paradigm from which food safety and public health practices arise. This shift is crucial in Africa, where traditional approaches to foodborne surveillance often fail to account for the interconnectedness of social, economic, and environmental factors[91]. By focusing on a systems thinking approach, we can better understand the complexities and multiple dimensions of public health challenges. This approach not only highlights the direct and indirect effects of food safety practices but also highlights the importance of integrating local knowledge and cultural practices into public health strategies. Enabling this paradigm shift involves broader educational initiatives, enhancing community engagement, and reforming policies to support a more holistic understanding of foodborne disease dynamics. Such a transformation can pave the way for more effective, sustainable, and culturally appropriate interventions that are critical for improving food safety and overall public health outcomes in these regions.

Based on the identified leverage points within the FBD surveillance system, the deep leverage points were found to hold the greatest potential for driving meaningful and transformative change within the system. To enhance system resilience and effectiveness, we propose the following policy recommendations based on the scenario analysis.

First, efforts to build and maintain public trust should be prioritized, since it will improve compliance and cooperation across sectors. Any policies implemented should be transparent and aim to engage local communities to strengthen trust in FBD surveillance.

Next, robust reporting systems should be implemented in order to ensure that safe food practices are followed, and governmental workers or

**Table 1 | Detailed description of each scenario developed in the scenario planning and implementation phase, as well as their impact on the leverage points as well as effectiveness, adaptability and resilience criteria**

| Scenario | Public trust | Compliance with food safety practices | Data sharing | Effectiveness | Adaptability | Resilience |
|---|---|---|---|---|---|---|
| Scenario 1: High trust and compliance due to transparency and strong food safety enforcement; limited data sharing due to inadequate infrastructure. | High | High | Limited | Effective in reducing FBD incidence, but limited by slower response times due to poor data sharing. | Limited adaptability during outbreaks due to slow response. | Moderately resilient due to strong compliance, but can increase outbreak risks due to delayed response. |
| Scenario 2: Low public trust due to high incidence, reducing community engagement; compliance remains high with strict regulations, high data sharing allows for effective coordination. | Low | High | High | Limited effectiveness; low trust reduces reporting and compliance with regulations. | Highly adaptable, with fast response times due to data sharing. | Moderately resilient due to compliance, but vulnerable to public disengagement. |
| Scenario 3: Stable but moderate trust and compliance, high data sharing through community efforts. | Moderate | Moderate | High | Moderately effective; limited compliance and public engagement, which reduce overall impact on FBD incidence. | Highly adaptable, with fast response times due to data sharing. | Moderately resilient, but there will be some persistence of FBD incidence due to low compliance. |
| Scenario 4: High trust driven by transparent outreach; low compliance, especially in informal food sectors, due to reduced enforcement; limited data sharing. | High | Low | Limited | Low effectiveness; contamination risks remains due to limited compliance and delayed response. | Limited adaptability due to slow response to outbreaks and low compliance. | Low resilience, as high trust alone cannot overcome poor compliance and data sharing. |
| Scenario 5: Ideal conditions with high levels of trust, compliance and data sharing, supported by strong regulations from health authorities and legislators; investment in infrastructure for data sharing. | High | High | High | Most effective; optimally reduces FBD incidence through strong compliance, trust, and response. However, may strain resources and lead to disruption if not sustained. | Highly adaptable due to strong collaboration and data sharing between sectors. Complex coordination requirements may lead to delays if collaboration is not sustained. | Resilient overall, but limited by infrastructure challenges. High data-sharing capacity requires reliable resources such as IT infrastructure, which may be lacking in many African countries. |

legislators should be allowed to intercept when procedures are not followed. This requires formal cross-sectorial collaboration among the local and/or regional public health, food safety, and environmental health authorities.

Finally, promotion of data and information sharing and improve education on food handling and preparation to avoid cross-contamination and improve hygiene practices should be prioritized. Protocols should be established for data exchange between different sectors, so that the surveillance system can be improved.

In the short term, efforts should focus on immediate improvements to compliance and data-sharing practices. Implementing low-cost, high-impact interventions such as compliance training for food handlers and encouraging data-sharing in surveys and projects can provide quick results. Immediate, targeted public trust initiatives, such as local health campaigns and community involvement in food safety initiatives, can also have an impact on improving the system's effectiveness.

For sustained impact, long-term strategies should focus on strengthening institutional frameworks for data sharing and compliance. This includes building digital infrastructure that aims to include data-sharing into policies and develop educational programs focused on food safety from early stages. Furthermore, long-term strategies should integrate continuous trust-building through transparency in public health interventions, sustained community engagement, and feedback mechanisms that allow public input on FBD policies.

To ensure a sustainable transition from the current FBD surveillance system to an improved one, it is essential to implement changes at all levels. Engaging local communities is crucial for identifying and reporting FBD outbreaks and illnesses. Without their support, it becomes challenging to trace the source of contaminated food and track those affected. Providing training and educational resources is beneficial, but the system's effectiveness relies on community willingness to participate and sustain these changes. Although such issues are typically addressed through a top-down approach, employing a bottom-up approach in this context can be advantageous as it emphasizes community involvement and decentralizes decision-making. Improving these systems dynamics can also help meet the UN Sustainable Development Goals (SDG), specifically 2 (Zero hunger) and SDG 3 (Good health and well-being)[92].

The focus of our analysis was to describe the overall drivers at the population level and, therefore, cannot be used to predict the quantitative effects of interventions. While we utilized knowledge from the literature to support the elements identified in the workshop, some drivers, such as food fraud and shopping habits, appear to be less researched than others. This discrepancy introduces a risk of certain elements in the system being more detailed than others.

In addition, there is a risk of personal bias from the authors when interpreting the elements identified in the stakeholder workshop. To mitigate this, we corroborated the elements with existing literature and only included those supported by other studies that concurred on the subject.

There is also a risk of generalization. This study focuses on Africa, which encompass a diverse range of countries, territories, and sub-communities. Consequently, some elements may have been simplified, potentially overlooking regional and cultural differences. However, we argue that this generalization was necessary to highlight broad trends that should be addressed. Further research should be conducted in the future to identify specific elements that need to be modified in the respective countries' FBD surveillance systems. Nevertheless, addressing the issue of FBD surveillance in Africa in this way can be useful to predict the relative effect of different interventions, and is considered a complementary approach to statistical or analytic thinking[93]. Further research is needed so that more detailed and context-specific data can be gathered, allowing for the development of tailored interventions that accurately address the unique challenges faced by different regions and communities in Africa.

## Conclusions
This study offers a comprehensive, systems-thinking-based examination of the challenges facing FBD surveillance in Africa, combining stakeholder insights with causal modelling, leverage point analysis, and scenario planning. Among the scenarios assessed, Scenario 5, characterized by high levels of public trust, compliance, and data sharing, emerged as the optimal strategy for strengthening surveillance system effectiveness. Achieving this scenario will require targeted investments in infrastructure, regulatory support, and public engagement.

By identifying structural and behavioral barriers and explore where strategic interventions might have the greatest impact, we provide a starting point for strengthening surveillance systems in resource-limited settings. Deep leverage points within the system offer the greatest potential for long-term improvement. To enhance surveillance effectiveness, efforts should focus on building public trust through transparency and community involvement, strengthening enforcement of food safety standards, and promoting cross-sectoral data sharing alongside improved education on hygiene and good handling. Short-term actions should target practical, low-cost interventions, while long-term strategies must include trust-building and compliance into institutional frameworks. These combined efforts provide a roadmap toward a more resilient and effective FBD surveillance system in resource-limited settings.

## Data availability
The original research presented in this study is included in the article or in the Supplementary Information. The list of participants in the expert workshop can be found in Supplementary Data 1. The results of the scoping review are available in Supplementary Data 2 and the Supplementary References. Further inquiries can be directed to the corresponding author.

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

## Acknowledgements

This study is part of the "FOCAL (Foodborne disease epidemiology, surveillance, and Control in African LMIC)" Project, a multi-partner, multi-study research grant co-funded by the Bill & Melinda Gates Foundation and

the Foreign, Commonwealth & Development Office (FCDO) of the United Kingdom Government (Grant Agreement Investment ID OPP1195617). The funder of the study had no role in study design, data collection, data analysis, data interpretation.

## Author contributions

TH conceived the idea for the study. C.T. conducted the literature review and generated the systems maps, with assistance from O.T. All authors contributed to the expert workshop and assessed the problem using the Iceberg model. T.H. and O.T. supported the work and reviewed the conceptualized feedback loops. C.T. wrote the manuscript, with help from O.T. All authors edited the manuscript.

## Competing interests

The authors declare no competing interests.
