## [Transparent Peer Review file · Communications Medicine]

Applying a systems thinking approach to evaluating the effectiveness of Africa's foodborne disease surveillance systems

Corresponding Author: Ms Cecilie Thystrup

Version 0:

Reviewer comments:

Reviewer #1

(Remarks to the Author)

Foodborne disease is a major public health concern and African regions have the highest per capita burden so this paper is timely and deals with an important issue.

There are some challenges.

- Much more information is needed on the stakeholders who developed the map. The merits of the analysis is very dependent on their expertise and accuracy but no information is given on this. What is their expertise in FBD surveillance? Their age, background and gender? Their training and background in causal theory? Why should we trust their analysis?
- Justification for why the "Iceberg" model is a suitable construct for this problem. Why was it chosen? What alternatives were considered? Has it been found useful in comparable situations?
- It is not clear whether the focus is FBD or FBD surveillance. Three of the four ice-berg questions do not refer to surveillance.

There is no validation or triangulation of the analysis. Consider the following

1. Internal validation:

- Coherence analysis: The internal logic of the model results is evaluated, verifying that the concepts are well defined, the relationships between them are clear and there are no contradictions.
- Analysis of exhaustiveness: It is verified that the model results addresses all relevant aspects of the research problem and does not omit important elements.
- Analysis of parsimony: It is evaluated whether the model results is simple and concise enough to be manageable, without sacrificing its ability to explain the phenomenon under study.

2. External validation:

- Triangulation of methods: Different research methods are used to collect data on the phenomenon under study and it is verified whether the results converge with the predictions of the model results.
- Expert feedback: Consult experts on the research topic to obtain their opinion on the validity of the model results. Delphi often used for this.
- Empirical tests: The hypotheses derived from the model results are tested through data collection and analysis.

Results

- The factors identified read as a "laundry list" of miscellaneous challenges.
- They are not clearly prioritised or linked.
- What are the justifications for inclusions of factors? What is omitted?
- The definitions and differences between different factors is not obvious. Why is "improper food handling" classified as "underlying structure" and "lack of prioritisation" a "pattern or trend"? Why is "poor cost-benefit" a "mental model".
- Likewise the logic behind the identification and choice of leverage points seems to lack coherence and systematic approach.

Other more minor points:

- Insufficient review of literature on previous management of FBD in Africa which goes far beyond management training and risk factor identification.
- Inconsistencies in capitalisation and grammar.

Reviewer #2

(Remarks to the Author)

Thank you for the opportunity to review this paper. It uses a combination of systems thinking tools and methods to understand food borne disease surveillance from several stakeholders across LMIC's in Africa. A strength of the paper is the use of a variety of systems thinking methods and the setting in which the paper took place. However, the paper has serious limitations that need to be addressed in order to make a contribution. These are listed in detail below, but mainly involve being clearer about your theories, definitions, and methods and transparent about how these were actually applied in your workshop.

Title

1. The title of your paper emphasises the iceberg model, but your results are a causal loop diagram – the title should be updated to reflect which systems thinking methods you used.

Abstract

2. Line 35: 'leverage points analysis' is unclear. This should be updated to more specifically reflect which method you used (e.g., coding against the iceberg, analysing a CLD).

3. Your methods don't make mention of system dynamics / feedback / causal loop diagrams, but your results are framed in terms of these (as opposed to in terms of the iceberg model), so your use of system dynamics / causal loop diagrams ought to be mentioned in the title.

4. Line 39: It would be helpful to be clearer about what a 'deep' leverage point means here – is it relative to the iceberg model or the CLD or something else?

Introduction

5. Lines 98 – 99: The use of 'Systemic thinking' followed by 'Systems thinking' is confusing as these are slightly different terms. I would either be consistent in the use of terms or explain (briefly) what the difference is.

6. Lines 98 – 101: I think that your definition of systemic thinking / systems thinking needs to be adjusted. The way you describe it comes across as a reductionist approach (i.e., simplifying a system and breaking it down into its parts to understand it) whereas systems thinking tries to understand how parts of a system interact to achieve a purpose, putting an emphasis on multiple parts, relationships, behaviour over time, and purpose.

7. Lines 105 – 107: System dynamics is one approach to systems thinking, but there's a need to explain what the relationship is between feedback loops and system dynamics, along with providing a broader definition of system dynamics.

8. Lines 108 – 109: With the addition of 'wicked problems' you have now introduced several different terms to describe systems thinking: systemic thinking, systems thinking, system dynamics, wicked problems. I suggest simplifying and streamlining this section to set the reader up to understand how you used systems thinking to guide your study. If CLDs and the iceberg model were the main tools you used, I would be setting up the explanation to reflect that.

9. Line 125: the Iceberg model needs to be defined and referenced. You could move your description of the iceberg model in lines 154 – 166 earlier or you could hold off on referencing the iceberg model until that point in the paper or you could signal that the iceberg model will be addressed in more detail later.

Methods

10. Line 137: Figure 1 introduces several terms / concepts that you haven't yet explained to the reader (e.g., iceberg model, system elements and boundaries, feedback loops, leverage analysis, deep leverage points, shallow leverage points). These should all be clearly defined before the figure so the reader can get a better understanding of the flow of your study.

11. Lines 138 – 140: What is the relationship between "Five-Phase Process of Systems Thinking and Modeling" and your figure? It would be helpful to understand if you followed this exactly or adapted it and combined it with something else.

12. Lines 140 – 148: I recommend having the list of participants included in the article rather than the supplement to make it really clear who contributed to this study.

13. Lines 172 – 185: More detail is needed to understand how these questions were actually applied. Were they discussed in a workshop? Interviews? How were the ideas analysed? Were they shared back with participants?

14. Lines 187 – 219: How did you actually do this with participants, and how did you verify their contributions? What is the justification for the link from the iceberg model to the system elements to CLD?

15. Lines 221 – 231: It's helpful here to understand how leverage points were defined and used, but again it needs to be made clear how these were explored with participants.

Results

16. Lines 234 – 265: your methods do not set up how you arrived at these results, particularly as the methods do not mention how you classified the different elements that came up or how you validated these with participants.

17. Lines 267 – 342: This is a rich description of what may be going on the system, but it's still not clear whether this came from the participants, the literature, or both. Additionally, the relationship between the diagram and the description is unclear. You should be starting with a reference mode: a change over time related to FBD, and describing your CLD as impacting the change over time, and your description should be organised around that.

18. Lines 348 – 387: This is again a rich and interesting description of the system, but Meadows only provides a list of leverage points, not a process to classify them. How was this carried out?

Reviewer #3

(Remarks to the Author)

SUMMARY OF MANUSCRIPT

1. The authors sought to investigate the underlying behaviours and challenges of FBD surveillance systems in African LMICs and their impact on their economies using the "iceberg" model and a system thinking approach. The authors relied on data from a food safety and public health stakeholder workshop held in Tanzania. The authors concluded that "transforming the underlying mindsets and educational frameworks and integrating culturally sensitive practices" will enhance FBD surveillance systems.

GENERAL COMMENT:

2. The title of the manuscript does not reflect its content. The authors, in their results, do not show a clear linkage between foodborne diseases and their impact on African economies. The authors use the words "hidden impact", I expected to see a causal relationship between foodborne diseases and African economies. The authors should consider revising the title to reflect FBD surveillance systems and replace the word "impact" which connotes a causal linkage (based on the available data, causal linkage cannot be demonstrated in this study).

3. It is unclear whether the authors assess foodborne diseases or foodborne disease surveillance systems. There was not a clear focus on either of them.

4. The authors mentioned the study covered 4 countries (Tanzania, Ethiopia, Mozambique, and Nigeria) but did not provide any country level analysis of the similarities and differences across countries.

5. In summary, the authors do not adequately address the objective of their study.

SPECIFIC COMMENTS

ABSTRACT

6. The abstract can be improved. The background does not highlight the problem with FBD surveillance, which is the focus of the study. The background focused on the general consequences and complexities of FBDs, not FBD surveillance systems.

PLAIN LANGUAGE SUMMARY

7. Similar concerns raised in the abstract above.

INTRODUCTION

8. Define "FBDs surveillance systems" when it is first introduced. It is the focus of the research and consider introducing it in earlier paragraphs and providing further details. It is currently the last but one paragraph.

9. Lines 123-125, the objective of the study does not reflect the title of the manuscript. Are you focusing on FBD surveillance systems or FBDs?

10. General comment: I suggest you restrict the paragraph on systems thinking to one paragraph and expand on the challenges and gaps in FBD surveillance in Africa. The background does not clearly communicate the challenges and gaps with the FBD surveillance system, making it a critical issue to study.

METHODS

11. The STUDY AREA is not described. Under this subheading, the authors can describe briefly the FBD surveillance system in the selected countries, provide a brief overview (1-3 lines) of the FBD issues in the countries.

12. The STUDY DESIGN is not discussed. The authors should describe briefly how the workshop was organised, type of participants and how data was collected. Show a pathway/process from gathering information in the workshop to data analysis. The study design should show that a multidisciplinary approach was adopted to address a "wicked problem" of FBD surveillance systems.

13. The STUDY DATA is not discussed. Lines 180-185 does not sufficiently describe how the data was collected and the nature of the data collected. Did the authors transcribe the workshop proceedings, took summary notes, basic profile of participants, professional background/expertise and how they potentially affect their data?

14. Line 136, the Supplementary Table S1 does not demonstrate "... the problem of FBD-surveillance in Africa LMICs through systems thinking". Table S1 contains the list of participants and their roles in the project.

15. Line 143, state the criteria for selecting these four countries out of the many countries in Africa. Did the authors use purposive sampling or random sampling? There should be a clear selection criterion of the four countries. It should be mentioned in the METHODS.

16. In Line 227, are you referring to FBD surveillance system or FBD system (Line 197). The two terms are different and contain different elements.

RESULTS

17. Line 236, the question asked was "... what is currently happening with foodborne diseases in Africa....". This is not the same as identifying what is happening with FBD-surveillance systems in LMIC. Did the former question sufficiently capture

what is currently happening with FBD-surveillance systems in LMIC?

18. Line 348, food safety is a public good and thus, requires a regulatory framework to ensure compliance. Government and its institutions are critical actors in ensuring a robust food safety and surveillance system. This is clearly missing from your results and discussion.

DISCUSSION

19. Lines 460-461, sentence is either missing or incomplete.

20. Line 490-497, your results have not demonstrated the lack of these recommendations in the current FBD surveillance systems. The recommendations you make must flow from your results and conclusions.

CONCLUSION

21. Line 536, you have not demonstrated in the current study the paradigms that govern public health strategies, which will necessitate a shift. The conclusion must flow from your results.

Version 1:

Reviewer comments:

Reviewer #1

(Remarks to the Author)

The points raised have been largely addressed

Reviewer #2

(Remarks to the Author)

Thank you for putting so much work into this paper and being so responsive to all reviewer comments. I can see the huge amount of work that went into revising this draft, and I think this is a significant step forward.

However, I have three remaining overarching concerns that I go into detail on below:

1. I think the paper needs to be significantly revised structurally to make it more concise and also make sure it flows well. In particular, large portions of the introduction should be combined with the methods section and/or eliminated.

2. This version of the paper has added extensive discussion of dynamic modelling. It's great you've explored this step, but I think this paper is doing too much, and it means that it's still difficult to see how all of your analyses come together to justify your conclusions. My overall recommendation is to remove dynamic modelling from this paper and suggest it as future research, and then work on the dynamic modelling you've done already as a follow up paper, allowing full explanation of the dynamic modelling.

3. The description of how you built your CLDs and what the CLDs told you is much improved in this paper. However, the figures in the Appendix do not seem to follow the conventions of system dynamics / match up well with the text – this needs to be addressed to ensure the paper all aligns.

Introduction

1. Lines 114 - 134– The introduction to systems thinking is quite abrupt, and possibly not justified. However, the description of 'wicked problems' is nice – I suggest removing the reference to systems thinking in lines 114 – 118 (possibly just remove these lines completely), and update lines 132 – 134 to explicitly connect the fact that wicked problems can be addressed using systems thinking.

2. Lines 141 – 145: The definition of system dynamics needs updating. Causal loops are structures that cause system dynamics (patterns of change over time).

3. Lines 149 – 296: I can see all the work you all put in to explain your various methods. Thank you for that. I do think now that the paper has swung too much in the other direction and has begun too long. I have made comments about lines 149 – 296, but after having read the methods, I've swung back here, and overarching, my recommendation is to combine lines 149 – 296 with the methods, and streamline.

4. Lines 149 - 152: These lines serve as a transition to wrap up your introduction and start your methods – I suggest doing this clearly and intentionally. Have a clear ending to your intro where you state the aims of your study, and then start a new section as methods. I recognise that you have a methods section later, but this part feels like a methods section more than an introduction.

5. Lines 155 – 159: Need a brief definition of the iceberg model here.

6. Line 168: The term 'This organised list' is confusing – needs to be clarified.

7. Line 172: I'm not familiar with 'analysis of exhaustiveness' – but this doesn't seem like the right description of what you did. Do you have a citation for the validation step that you did? If not, that's okay, but then in that case, in lines 174-175, I suggest being clearer about what you 'cross-checked' against.

8. Line 176: Is 'Archetype detection' an official process? If so, it needs a citation. Either way, a clearer description of what this step involved is needed.

9. Line 181: Did you have a better understanding of the system dynamics or the system structure or both?

10. Lines 184 – 186: dynamic modelling is a complex process – more detail is needed in how you actually carried that out.

11. Lines 187 – 194: This process needs further description and possibly citations.

12. Lines 207 – 208: Is the term 'structural' needed there given the rest of the list?

13. Line 209: I suggest replacing 'the mental structures that people hold' with 'People's deeply held beliefs'

14. Lines 204 – 296: Thank you for the comprehensive description of all the different theories / tools you used. However, I note that this manuscript has become quite long (57 pages) and would suggest that it may be beneficial to significantly

reduce the length of these sections to instead preference more details about what you actually did in the study (in the methods section).

Methods

15. Lines 335 – 342: Acknowledging the paper is already quite long, I think you need a (short) supplement section outlining your search strategy for the scoping review + inclusion / exclusion criteria. Updating having read further: I think you have done this in lines 511 – 518, which I recommend moving to the methods section. I recommend having the first two columns (Concept & Search string) in the methods and the second 2 columns (Number of hits & relevant publications) in the results. Alternatively, you could leave all of Table 2 in the results, and just briefly note in the methods that some of these details are in Table 2 in the results.

16. Lines 349 – 350: The use of centrality in CLDs has been critiqued – you’re already doing so much in this paper, I recommend removing that portion from your methods / analysis. Additionally, you don’t seem to pick this up again in the results, which is even more reason to just remove this line.

17. Lines 385 – 445: These lines read more as results. Additionally, the equations used here could probably go in an appendix (it’s great you’ve included them!), with just a high level summary of the model + the results of the simulations in the results section of the paper. I would also encourage you to consider removing the dynamic modelling element of this paper, spend more time on finalising your dynamic modelling work, and publish it later as a separate paper. The paper is doing too much (although it’s all great work!) and the dynamic modelling part deserves its own breathing space and room to fully explain.

18. Lines 462 – 467: needs citations and/or more details

19. Lines 468 – 471: great that you did this but need more detail – was this in a focus group, interview, workshop, discussion? How many experts?

20. Lines 525 – 573: These are really nice explanations of your loops, thank you. However, in the supplement / overall map, there are some issues that need to be addressed (see comments on supplement below).

21. Table 3: I’d label the first column as ‘loops,’ not CLDs.

22. Line 655 – 695: Having now read the results here, I feel even more strongly that the dynamic modelling portion should be removed and saved for a future paper. That will give the remaining elements the breathing room and clarity, and also give you the opportunity to report on the dynamic modelling in more detail in a separate publication.

23. Lines 696 – 750: Building on my points on dynamic modelling, if you decide to remove that, I think you could clarify and reduce this section. I’m not clear on if this was further run through the simulation model or whether the comments / table were developed in a different way. If these are further simulation model results, they would require even more explanation, adding even more length to the paper. My advice therefore would be to write this section just based on what arose from the stakeholder engagement / literature review / CLD / leverage point analysis, and then in your discussion, point to dynamic modelling as the next step to develop this further.

Discussion

24. Similarly to above, while your discussion makes reference to dynamic modelling, it appears the bulk of your discussion comes from all the other steps (besides dynamic modelling) further justifying removing dynamic modelling from this paper and having it as a future paper.

Supplementary Figure S4 – S11

25. I have a significant amount of experience with building causal loop diagrams, and I found the loops in these figures very hard to interpret. I couldn’t figure out which polarity (positive or negative) the red and blue coincided with, and the ‘double negatives’ that arose from variables like “Inaccurate and incomplete diagnosis”; “Ill-equipped diagnostic facilities”; “Lack of education”; “Underreporting of disease”; “Mistrust in health interventions” compounded this confusion. I found the descriptions of the loops in the main text helpful and meaningful. Reviewing the diagrams, I’m not certain that they align with the conventions of system dynamics and reinforcing and balancing loops, creating uncertainty and confusion about your analysis, including into the archetype and dynamic modelling phase. I think it would be great to re-do these CLDs so that they closely follow the conventions of SD and match the text. This effort would also be more important, in my opinion, than expanding the dynamic modelling section, and is further justification to hold over dynamic modelling for a future paper.

Reviewer #3

(Remarks to the Author)

GENERAL COMMENT

The authors have done a good job revising the manuscript. They have addressed many of the comments, and the current version is a significant improvement on the earlier version. However, further revision is needed to improve the manuscript.

TITLE

Unfortunately, the new title of the manuscript does not reflect its content. “...combat inequality...” inequality only appears in the title. The authors do not mention or discuss inequality anywhere in the manuscript except in the title. Based on the content of the manuscript, I suggest a title along the lines of [The Effectiveness of Africa’s Foodborne Disease Surveillance Systems: A Systems Thinking and Modelling Approach]

INTRODUCTION

Line 81, “...tends” should be [trends]

Line 90, “locally consumed food”, what does it mean? It is confusing. It does imply that then there is internationally consumed food. Locally produced food (line 94) is understandable but not locally consumed food.

Line 103, “...ignorance of food-safety practices;”, This is an inaccurate statement. There is enough literature on food safety showing that African consumers are generally knowledgeable about food safety. I suggest you either delete or consider replacing it with [low compliance with food-safety practices]

Line 105, "antimicrobial resistance", I suggest introducing the abbreviation [AMR] before using just AMR in line 107.
Lines 152-296, all this section should be moved under the METHODS sections. Refer to the journal's manuscript style and formatting guide

General comment: I don't see the research question clearly stated in the introduction. Line 149-152, "...systematically explore and address the challenges hindering the implementation and effectiveness of FBD surveillance systems in African LMICs". If this is the research question/statement, I suggest you rephrase it appropriately and clearly. It is currently not clear enough.

METHODS

I suggest you move Lines 152-296 under methods. If you think that will make it long, then move some sections to supplementary materials.

DISCUSSION

I don't see where the discussion ends and the conclusion starts.

CONCLUSION

Where does the conclusion start? There is no conclusion heading.

SUPPLEMENTARY MATERIAL

Supplementary Table S1, I will not include the names of the workshop participants for confidentiality reasons and pseudonymization.

Version 2:

Reviewer comments:

Reviewer #3

(Remarks to the Author)

GENERAL COMMENT

Once again, I thank the authors for taking on board the suggestions and their commitment to improve the manuscript. In future responses, I suggest you provide line numbers to sections you have effected changes in your rebuttal table. Going by just the track changes in the manuscript slows down the review process.

TITLE

Good

ABSTRACT

Line 23; African LMICs? Delete LMICs. Except for Seychelles, no African country is a high-income country. They are either low or middle income countries. Therefore, just write "while previous efforts in Africa have..."

Line 31; African LMICs again. Correct accordingly throughout the manuscript.

INTRODUCTION

Line 101-105, sentence is too long (52 words). Rephrase it.

Line 105-108, sentence is too long (41 words). Rephrase it.

General comment: My concern about the research question has not been addressed. The aim of the study is not the same as the research question. The research question must be concise. Example, how do structural, behavioural and contextual barriers limit an effective FBD surveillance system in Africa? A question framed this way allows you to identify the barriers and analyse their pathways to influence/limit effective FBD surveillance systems. That's what I hope to see from a research question. So, once again, I suggest you frame your research question to reflect your work.

METHOD

Line 257, ...structural, behavioral, or contextual challenges, provide an explanation or a definition of these classifications. Providing very brief definitions will assist the reader know the boundaries within which you are using these words.

RESULTS

Line 325-326, additional literature...outside scoping review, why will this relevant literature be outside the scoping review? If it is relevant, it should have popped up in the scoping review?

Line 329-331, I'm confused here. What is the difference between literature from the scoping review and complementary literature? If you are finding relevant literature (complementary literature) outside your scoping review, will you not be suffering from missing data bias? I may not be understanding what you mean by complementary literature. Your restriction to only PubMed may account for why you have so many complementary literatures. Be mindful of the effect on your results if you were to include other databases.

Line 365, "...internal, external, or outside elements..." stick to one word, either use external or outside. Line 271-272, be consistent with the use of inside or outside OR internal or external.

Table 2, education and cost-benefit..., "The underlying...lack of food safety awareness and education...". Factually inaccurate statement. You can use inadequate/limited/low food safety awareness... or similar words.

Line 477, "lack of education", similar comment as above. There is enough literature to show households and food handlers in Africa have basic food safety knowledge and awareness, but it may not be enough to sufficiently mitigate foodborne

disease risks. To be on the safe side use insufficient/inadequate or similar synonyms.
Make the needed changes in other parts of the manuscript where you argue “lack of education”
Line 520, “...changed”, do you mean “change”?

DISCUSSION

Line 565-570, the discussion should be about your results and not rearguing parts of your introduction. This paragraph is not necessarily discussing any part of your results.

Line 579, “...handler” should be “handlers”

Line 590, “... animals to produce”. For better clarity, I would change produce to crops or plants.

Line 608, “...FBD surveillance...” I suggest you add “by governments” so, A lack of prioritization of FBD surveillance by governments...

Line 639-641, I suggest you provide a reference.

Line 653-654, I suggest you delete “these recommendations...analysis”

Line 651-668, I suggest recommendations form part of the conclusion.

CONCLUSION

The conclusion is incomplete. The “take away” message or novel finding is missing. I suggest you incorporate Lines 40-41 and Lines 46-47 in the conclusion and add the recommendations.

Comments on rebuttal to comments of reviewer #1

GENERAL: I read the comments of reviewer #1 and the responses of the authors. Then, I verified the changes made by the authors in the text. Many of the comments raised by the reviewer have been satisfactorily addressed. However, a few comments will require further minor action.

Feedback on the rebuttal table: In the future, the authors should indicate where (new line numbers) the changes have been made.

COMMENTS: The authors have satisfactorily addressed many of the suggestions, reflecting the improved status of the manuscript. The authors should take further action on the following few issues.

1. Line 159, iceberg definition or description was provided without an appropriate reference. A reference was provided in line 227.
2. Line 201, “...analysis of parsimony...”, provide appropriate reference.
3. Line 202, “...analysis of exhaustiveness...”, provide appropriate reference.
4. Line 235, correct spelling of “... deeply held beliefs...”
5. Table 1 (Search strategy for scoping review...), it is too long (8 pages of a single table?). I suggest it will be appropriate if the whole table is moved to supplementary material and you can keep the first and last columns in the main text.
6. Line 311-313, the revised sentence is still not clear. Were the scenarios presented DURING the workshop or were they sent to the participants for comments AFTER the workshop?
7. Supplementary Figures S4–S11 satisfactorily addressed.

	Reviewer comment	Response
	Reviewer #1	
1	Foodborne disease is a major public health concern and African regions have the highest per capita burden so this paper is timely and deals with an important issue.	We agree with this observation, and we think that improving the paper will ensure that the message of the paper shines through more clearly.
2	Much more information is needed on the stakeholders who developed the map. The merits of the analysis is very dependent on their expertise and accuracy but no information is given on this. What is their expertise in FBD surveillance? Their age, background and gender? Their training and background in causal theory? Why should we trust their analysis?	The participants who participated in the workshop were individuals involved in a broader, ongoing study across multiple African countries, selected for their experience with foodborne diseases in African countries. These participants were chosen because of their diverse backgrounds and first-hand experience with foodborne diseases in Africa. Their backgrounds included doctors and health workers, government officials responsible for food safety in their respective countries, and researchers with significant field experience in food safety and public health. Prior to the problem-structuring phase, participants received targeted training in causal thinking to enhance their ability to contribute meaningfully to the systems analysis. To further validate their experiences, we supported their findings with a literature review, confirming alignment between their observations and documented trends in the field. Given their expertise and the preparatory training, we are confident that their contributions provided a solid and trustworthy foundation for identifying key challenges in implementing and improving FBD surveillance in this context. We have emphasized this in the section of "Applying the Iceberg model in a workshop" (line 300-333)
3	Justification for why the "Iceberg" model is a suitable construct for this	The "Iceberg" model has been used The Iceberg model was chosen

	problem. Why was it chosen? What alternatives were considered? Has it been found useful in comparable situations?	primarily because it has been successfully applied in similar public health contexts, such as antimicrobial resistance and outbreak response (line 146-147), and is widely recognized as an effective tool for problem structuring. We thought that integrating the Iceberg model within the Five-Phase Systems Thinking methodology would provide a robust framework for identifying root causes and mapping complex challenges in FBD surveillance.
4	It is not clear whether the focus is FBD or FBD surveillance. Three of the four ice-berg questions do not refer to surveillance.	We have clarified in the manuscript that the focus was FBD surveillance and not FBDs in general. We appreciate the reviewers' comments on the clarity of the workshop questions. Prior to the discussion, workshop participants received a comprehensive introduction to the primary goal of the study: identifying the challenges hindering the implementation and effectiveness of FBD surveillance in African LMICs. This briefing ensured that participants understood the focus on surveillance systems, even though some of the Iceberg questions themselves did not directly reference surveillance. We believe that this context sufficiently oriented participants to respond with an emphasis on FBD surveillance-related challenges.
5	There is no validation or triangulation of the analysis. Consider the following  1. Internal validation:  • Coherence analysis: The internal logic of the model results is evaluated, verifying that the concepts are well defined, the relationships between them are clear and there are no contradictions. • Analysis of exhaustiveness: It is verified that the model results addresses all relevant aspects of the research problem and does not omit important elements. 	We appreciate the reviewer's suggestion and we have added validation steps in each of the five phases of the Systems Thinking model. The specific validation method have been explained in the methodology and is shown in context on Figure 1.

	 • Analysis of parsimony: It is evaluated whether the model results is simple and concise enough to be manageable, without sacrificing its ability to explain the phenomenon under study. 2. External validation:  • Triangulation of methods: Different research methods are used to collect data on the phenomenon under study and it is verified whether the results converge with the predictions of the model results. • Expert feedback: Consult experts on the research topic to obtain their opinion on the validity of the model results. Delph often used for this. • Empirical tests: The hypotheses derived from the model results are tested through data collection and analysis. 	
6	The factors identified read as a “laundry list” of miscellaneous challenges.	The elements and behaviors identified in the workshop were grouped based on the level at which they had the largest impact (i.e. broader society, the healthcare system or the food- and agricultural area). In the new version of the manuscript, we included more detailed explanations of the elements, supporting them with results of the literature search.
7	They are not clearly prioritised or linked.	After rewriting the methodology and results section, it should be more clear to the reviewer how the factors are linked and prioritized. Specifically, we used the methodology to emphasize how we landed at the results.
8	What are the justifications for inclusions of factors? What is omitted?	The justifications for focusing on specific leverage points are detailed in the methodology and results sections. After constructing causal loop diagrams (CLDs), we conducted a leverage point analysis, which guided our attention toward the deeper, structural elements of the system that have the most potential for impactful change. This approach ensured that we concentrated on factors with the

		greatest influence on FBD surveillance outcomes, while other, less influential points were omitted to maintain clarity and focus. We hope that this is more clear in the revised version of the manuscript.
9	The definitions and differences between different factors is not obvious. Why is “improper food handling” classified as “underlying structure” and “lack of prioritisation” a “pattern or trend”? Why is “poor cost-benefit” a “mental model”.	We agree and we have added more information on how the different elements were classified in the “Iceberg model” section (line 204-236). To clarify the elements that the reviewer has highlighted, “improper food handling” was classified as an underlying structure because it refers to established practices that contribute to recurring foodborne risks. “Lack of prioritization” is categorized as a pattern/trend because it reflects a recurring behavior over time. “Poor cost-benefit perception” was classified as a mental model, because it represents a belief about the perceived economic viability of food safety practices in the system.
10	Likewise the logic behind the identification and choice of leverage points seems to lack coherence and systematic approach.	After reading the section on leverage points again, we agree with the reviewer that the logic behind why these leverage points were chosen were not clear enough. It should be more clear from the methodology now how we arrived at the leverage points in the section titled “Leverage point perspective and system dynamics” (line 360-376)
11	Insufficient review of literature on previous management of FBD in Africa which goes far beyond management training and risk factor identification.	In response to this comment, we revisited the literature on previous FBD management efforts in Africa and have expanded the introduction to include additional relevant information (see lines 86-117)). However, our literature review is intended as a supplementary background to contextualize the elements identified during the workshop, rather than as an exhaustive assessment of all FBD management literature.

12	Inconsistencies in capitalisation and grammar.	We have checked the manuscript for inconsistencies in capitalization and grammar in the new version.
Reviewer #2		
13	Thank you for the opportunity to review this paper. It uses a combination of systems thinking tools and methods to understand food borne disease surveillance from several stakeholders across LMIC's in Africa. A strength of the paper is the use of a variety of systems thinking methods and the setting in which the paper took place. However, the paper has serious limitations that need to be addressed in order to make a contribution. These are listed in detail below, but mainly involve being clearer about your theories, definitions, and methods and transparent about how these were actually applied in your workshop.	We thank the reviewer for this comment and we agree with the main points about the unstructured-ness of the paper. To point out a few changes, we went back to re-write the introduction and methodology sections to accurately reflect the systematic approach that we took to the problem of FBD surveillance in African LMIC. We also added a validation step for each level of the “Five Phase Process of Systems Thinking” which was the methodology we followed throughout the paper. In this new version of the analysis, we also added a dynamic modelling step, where we modelled the behavior of the leverage points to illustrate the points of the paper. It was a general request from all reviewers that we were more transparent about the workshop process, so we have added a section describing how these people were selected and how the workshop was conducted.
14	The title of your paper emphasises the iceberg model, but your results are a causal loop diagram – the title should be updated to reflect which systems thinking methods you used.	We changed the title to “Systems mapping of foodborne diseases in Africa to combat inequality in FBD surveillance systems” to accurately reflect the content of the paper and avoid the causal loop.
15	Line 35: ‘leverage points analysis’ is unclear. This should be updated to more specifically reflect which method you used (e.g., coding against the iceberg, analysing a CLD).	We have rephrased the introduction to describe how we conducted the leverage point analysis. This should give a more clear explanation of how we were able to detect leverage points. See line 182-186: “Specifically, we examined which variables were central to the reinforcing or balancing feedback loops and assessed which interventions that could potentially disrupt the negative patterns detected in the archetype analysis. To further validate these leverage points, we used dynamic modelling to simulate

		behavior-over-time graphs to assess how each leverage point would impact the system.”
16	Your methods don't make mention of system dynamics / feedback / causal loop diagrams, but your results are framed in terms of these (as opposed to in terms of the iceberg model), so your use of system dynamics / causal loop diagrams ought to be mentioned in the title.	The title has been rephrased to more accurately reflect the systems thinking approach, as has the methodology. The new title reads: “Systems mapping of foodborne diseases in Africa to combat inequality in FBD surveillance systems”
17	Line 39: It would be helpful to be clearer about what a ‘deep’ leverage point means here – is it relative to the iceberg model or the CLD or something else? Introduction	The concept of leverage points has been expanded on in the introduction, and the methodology to how they have been identified has also been more clearly phrased. Specifically in the introduction under subheading “Dynamic modelling and leverage point analysis” (line 266-285) and in the methodology section (line 370-444).
18	Lines 98 – 99: The use of ‘Systemic thinking’ followed by ‘Systems thinking’ is confusing as these are slightly different terms. I would either be consistent in the use of terms or explain (briefly) what the difference is.	We have changed the wording to only include systems thinking, which is the correct term to use in this setting.
19	Lines 98 – 101: I think that your definition of systemic thinking / systems thinking needs to be adjusted. The way you describe it comes across as a reductionist approach (i.e., simplifying a system and breaking it down into its parts to understand it) whereas systems thinking tries to understand how parts of a system interact to achieve a purpose, putting an emphasis on multiple parts, relationships, behaviour over time, and purpose.	We have rephrased the sentence to clarify the meaning of the systems thinking approach (line 133-146).
20	Lines 105 – 107: System dynamics is one approach to systems thinking, but there's a need to explain what the relationship is between feedback loops and system dynamics, along with providing a broader definition of system dynamics.	We have expanded the section on causal loop diagrams (CLDs) to clarify the relationship between feedback loops and system dynamics within the Five-Phase Systems Thinking model (line 239-264).
21	Lines 108 – 109: With the addition of ‘wicked problems’ you have now	This section has been completely rewritten to clarify the methodology of

	introduced several different terms to describe systems thinking: systemic thinking, systems thinking, system dynamics, wicked problems. I suggest simplifying and streamlining this section to set the reader up to understand how you used systems thinking to guide your study. If CLDs and the iceberg model were the main tools you used, I would be setting up the explanation to reflect that.	the paper. The terms and their role in systems thinking have been moved to the introduction under their respective subheading. The concept of 'wicked problems', meaning what a wicked problem is and why FBD surveillance is a 'wicked problem' has been moved to the introduction, line 120-132: "The problem of FBD surveillance can be described as a 'wicked problem', where the effects of the problem are interlinked in ways that involves contradictory and changing requirements (23–25). This means that there is no determinable place in the problem where a single solution can be made, thereby increasing the number of potential solutions (25). Such 'wicked problems' can have large, long-term effects on social, economic or environmental elements (25). They can also include time lags, where there will be a delay from the onset of the cause to the effect (25). In many African LMICs, the link between the contamination of food products and the observed increase in illnesses is rarely established due to the absence of effective surveillance systems (26,27). While the time lag between contamination and health outcomes may be shorter than in other complex health challenges, the inability to detect, attribute and monitor such events undermines both the recognition of the problem's scale and the ability to evaluate the effect of any intervention efforts (28). As a result, even effective prevention measures may go unnoticed, reducing the perceived incentive to invest in FBD control".
22	Line 125: the Iceberg model needs to be defined and referenced. You could move your description of the iceberg model in lines 154 – 166 earlier or you could hold off on referencing the iceberg model until that point in the	The 'Iceberg' model methodology has been moved to its own section in the introduction (line 205-236)

	paper or you could signal that the iceberg model will be addressed in more detail later.	
23	Line 137: Figure 1 introduces several terms / concepts that you haven't yet explained to the reader (e.g., iceberg model, system elements and boundaries, feedback loops, leverage analysis, deep leverage points, shallow leverage points). These should all be clearly defined before the figure so the reader can get a better understanding of the flow of your study.	The terms and their role in systems thinking have been moved to the introduction under their respective subheading. The concept of 'wicked problems', meaning what a wicked problem is and why FBD surveillance is a 'wicked problem' has been moved to the introduction (line 120-132).
24	Lines 138 – 140: What is the relationship between “Five-Phase Process of Systems Thinking and Modeling” and your figure? It would be helpful to understand if you followed this exactly or adapted it and combined it with something else.	We have completely changed the figure to more accurately disclose how we followed the “Five Phase Process of Systems Thinking and Modelling”. The new figure now shows what methods were used in each step, which corresponds to the individual phases of the model (see Figure 1).
25	Lines 140 – 148: I recommend having the list of participants included in the article rather than the supplement to make it really clear who contributed to this study.	Initially, we had moved the list of participants to Table 1 in the manuscript as per the reviewer's suggestion, but we feel that with the new version of the paper, the participant list fits better in the supplementary as Supplementary Table S1. We hope that the reviewer agrees.
26	Lines 172 – 185: More detail is needed to understand how these questions were actually applied. Were they discussed in a workshop? Interviews? How were the ideas analysed? Were they shared back with participants?	The questions were applied during a structured workshop involving key stakeholders with expertise in FBD surveillance. Participants engaged in facilitated group discussions to explore each question, allowing for collaborative identification of key elements and patterns. The insights gathered were systematically analyzed by organizing responses into themes and mapping them within the Iceberg model. Preliminary findings were shared back with participants for validation and refinement, ensuring that the analysis accurately reflected their perspectives and expertise. This is also clarified in line 320-332.

27	Lines 187 – 219: How did you actually do this with participants, and how did you verify their contributions? What is the justification for the link from the iceberg model to the system elements to CLD?	In the expert workshop, participants engaged in the problem structuring phase by addressing targeted questions based on the Iceberg model, which helped them identify events, patterns, underlying structures, and mental models related to FBD surveillance. Their insights were gathered through facilitated discussions and later organized into system elements. To verify their contributions, we conducted a validation step where preliminary findings were presented back to the participants for feedback and refinement. The transition from the Iceberg model to system elements and, subsequently, to CLDs was justified by the need to translate the qualitative insights into a systems framework. This approach enabled us to capture the dynamic interactions among elements, allowing for deeper analysis through system dynamics. The authors independently carried out the subsequent phases, leveraging the workshop insights as a foundation for developing CLDs and conducting the leverage point analysis.
28	Lines 221 – 231: It's helpful here to understand how leverage points were defined and used, but again it needs to be made clear how these were explored with participants.	After rewriting the methodology and results section, it should be more clear to the reviewer how the factors are linked and prioritized. Specifically, we used the methodology to emphasize how we landed at the results. Specifically, the section on "Leverage point perspective and system dynamics" in the methodology (line 360-369) and the results section titled "Leverage points of the FBD systems map" (lines 614-652) should clarify this.
29	Lines 234 – 265: your methods do not set up how you arrived at these results, particularly as the methods do not mention how you classified the different elements that came up or how you validated these with participants.	Again, the section on "Leverage point perspective and system dynamics" in the methodology (line 342-368) and the results section titled "Leverage points of the FBD systems map" (lines

		614-652) should clarify this much better.
30	Lines 267 – 342: This is a rich description of what may be going on the system, but it's still not clear whether this came from the participants, the literature, or both. Additionally, the relationship between the diagram and the description is unclear. You should be starting with a reference mode: a change over time related to FBD, and describing your CLD as impacting the change over time, and your description should be organised around that.	We have taken this comment into consideration and changed the section substantially. Now, a better explanation of the diagram has been added in the results section named "Causal loop diagrams and archetype definitions". Explanations of the CLDs identified in the overall map has been added which should help clarify how the CLDs could impact change over time. We hope that this has been more clear to the reviewers now.
31	Lines 348 – 387: This is again a rich and interesting description of the system, but Meadows only provides a list of leverage points, not a process to classify them. How was this carried out?	We agree that the methodology was not clear in terms of how we classified them. In the methodology, we added a section describing specifically how we used Meadows leverage point system to classify them (line 363-368): "Specifically, we focused on variables central to the reinforcing or balancing feedback loops, and assessed which interventions that could potentially disrupt the negative patterns detected in the archetype analysis. To further validate these leverage points, we used dynamic modelling to simulate behavior-over-time graphs to assess how each leverage point would impact the system. Different interventions that intend to improve FBD surveillance in LMICs and mitigate disease were listed and categorized, based on the level of impact of the leverage points."
	Reviewer #3	
32	The authors sought to investigate the underlying behaviours and challenges of FBD surveillance systems in African LMICs and their impact on their economies using the "Iceberg" model and a system thinking approach. The authors relied on data from a food safety and public health stakeholder workshop held in Tanzania. The authors concluded that "transforming the underlying mindsets and	We have noted the comment, and we have made significant changes in the manuscript.

	educational frameworks and integrating culturally sensitive practices" will enhance FBD surveillance systems.	
33	The title of the manuscript does not reflect its content. The authors, in their results, do not show a clear linkage between foodborne diseases and their impact on African economies. The authors use the words "hidden impact", I expected to see a causal relationship between foodborne diseases and African economies. The authors should consider revising the title to reflect FBD surveillance systems and replace the word "impact" which connotes a causal linkage (based on the available data, causal linkage cannot be demonstrated in this study).	We have changed the title to more accurately reflect the methodologies, and the causal relationships investigated in the study.
34	It is unclear whether the authors assess foodborne diseases or foodborne disease surveillance systems. There was not a clear focus on either of them.	We have changed the wording in the manuscript so that it is clear that the focus of the study was foodborne disease surveillance systems.
35	The authors mentioned the study covered 4 countries (Tanzania, Ethiopia, Mozambique, and Nigeria) but did not provide any country level analysis of the similarities and differences across countries.	Our study did not include a country-specific analysis of similarities and differences. Rather, the workshop aimed to draw on the diverse backgrounds and experiences of participants from Tanzania, Ethiopia, Mozambique, and Nigeria to collectively address the broader, cross-cutting challenges of FBD surveillance in African LMICs. In this way, we could provide a more comprehensive view of shared systemic issues relevant to the continent which were applicable across different settings.
36	In summary, the authors do not adequately address the objective of their study.	We thank the reviewer for this comment and we agree with the main points about the unstructured-ness of the paper. To point out a few changes, we went back to re-write the introduction and methodology sections to accurately reflect the systematic approach that we took to the problem of FBD surveillance in African LMIC. We also added a validation step for

		each level of the “Five Phase Process of Systems Thinking” which was the methodology we followed throughout the paper. In this new version of the analysis, we also added a dynamic modelling step, where we modelled the behavior of the leverage points to illustrate the points of the paper. It was a general request from all reviewers that we were more transparent about the workshop process, so we have added a section describing how these people were selected and how the workshop was conducted.
37	The abstract can be improved. The background does not highlight the problem with FBD surveillance, which is the focus of the study. The background focused on the general consequences and complexities of FBDs, not FBD surveillance systems.	We have changed the abstract completely to more accurately reflect the methodology and the overall goal of FBD surveillance.
38	Similar concerns raised in the abstract above.	We have noted the comment, and we have made significant changes in the manuscript.
39	Define “FBDs surveillance systems” when it is first introduced. It is the focus of the research and consider introducing it in earlier paragraphs and providing further details. It is currently the last but one paragraph.	We have defined the term “FBD surveillance system” in the introduction after the suggestion from the reviewer and clearly explained what elements we consider being a part of a FBD surveillance system: “Effective surveillance systems for FBDs are crucial for early detection, response and control of outbreaks and mitigation of risk (3,6). These systems can cover a variety of elements, such as routine laboratory testing, monitoring of specific pathogens, and the integration of advanced molecular technologies such as Whole Genome Sequencing (WGS) and metagenomics ((7,8)). For instance, many countries in the European Union (EU) have implemented robust surveillance systems for foodborne pathogens such as Campylobacter spp., and non-typhoidal Salmonella, which include systematic sampling and sharing of

		real-time data across multiple countries (9,10)." (line 74-80)
40	Lines 123-125, the objective of the study does not reflect the title of the manuscript. Are you focusing on FBD surveillance systems or FBDs?	We have changed the title of the manuscript to accurately reflect the focus of the manuscript, which is FBD surveillance systems.
41	General comment: I suggest you restrict the paragraph on systems thinking to one paragraph and expand on the challenges and gaps in FBD surveillance in Africa. The background does not clearly communicate the challenges and gaps with the FBD surveillance system, making it a critical issue to study.	We have changed the paragraph on systems thinking and expanded on the challenges and gaps in FBD surveillance in Africa (see line 90-114 for details).
42	The STUDY AREA is not described. Under this subheading, the authors can describe briefly the FBD surveillance system in the selected countries, provide a brief overview (1-3 lines) of the FBD issues in the countries.	We have clarified the study area in the introduction to emphasize that this paper aims to identify challenges relevant to FBD surveillance across African LMICs as a whole, rather than focusing on specific countries. Additionally, we have expanded the introduction to include further descriptions of FBD surveillance issues across multiple African countries, supported by relevant literature to provide context and background for the study's broader focus (see line 90-114 for details).
43	The SUDY DESIGN is not discussed. The authors should describe briefly how the workshop was organised, type of participants and how data was collected. Show a pathway/process from gathering information in the workshop to data analysis. The study design should show that a multidisciplinary approach was adopted to address a "wicked problem" of FBD surveillance systems.	We have expanded on the study design in the introduction section to provide a detailed overview of the workshop organization, participant backgrounds, and data collection process. The workshop brought together a diverse group of stakeholders, including healthcare professionals, government officials, and researchers, to capture a multidisciplinary perspective on FBD surveillance challenges. Information gathered in the workshop was systematically documented and analyzed using a structured pathway. This process involved identifying key elements through collaborative discussions, categorizing these elements within the

		“Iceberg” model, and using them to develop CLDs that highlight leverage points within the system.
44	The STUDY DATA is not discussed. Lines 180-185 does not sufficiently describe how the data was collected and the nature of the data collected. Did the authors transcribe the workshop proceedings, took summary notes, basic profile of participants, professional background/expertise and how they potentially affect their data?	We have expanded on how the data was collected in the methodology section: (line 323-332): “Workshop proceedings were documented by recording each group’s responses to the Iceberg model questions, with specific attention given to the different layers identified by each participant group. Summary notes were taken for each group’s iceberg model, capturing the key elements, patterns, structures, and mental models they identified. To address potential bias, we ensured that discussions were facilitated in a way that encouraged open contributions from all participants, reducing the influence of any single perspective on the collective output. Additionally, each group’s data was reviewed independently to identify any outlying perspectives and ensure consistency across responses. This approach allowed us to capture a balanced view of the system’s dynamics without any single participant’s background influencing the results.” Additionally, we have now provided a more detailed profile of participants, including their professional backgrounds and areas of expertise.
45	Line 136, the Supplementary Table S1 does not demonstrate “... the problem of FBD-surveillance in Africa LMICs through systems thinking”. Table S1 contains the list of participants and their roles in the project.	In light of the previous comments, the whole section has been rewritten in order to address the comments. The methodology explaining how the problem of FBD surveillance was addressed through system thinking should be clearer in the introduction as well as the methodology.
46	Line 143, state the criteria for selecting these four countries out of the many countries in Africa. Did the authors use purposive sampling or random sampling? There should be a clear selection criterion of the four countries.	The selection of participants from these four countries was not based on specific criteria for this workshop alone but rather on their involvement in a larger, ongoing research project across African LMICs. When countries were

	It should be mentioned in the METHODS.	initially selected for this broader project, we aimed to achieve geographical representation from southern, eastern, and western sub-Saharan Africa, ensuring diverse regional perspectives. Consequently, researchers from Tanzania, Ethiopia, Mozambique, and Nigeria were invited to participate in this workshop as part of their roles in the larger project.
47	In Line 227, are you referring to FBD surveillance system or FBD system (Line 197). The two terms are different and contain different elements.	We have clarified in the manuscript that the focus was FBD surveillance and not FBDs in general. The sentence referred to by the reviewer has been removed.
48	Line 236, the question asked was "... what is currently happening with foodborne diseases in Africa...". This is not the same as identifying what is happening with FBD-surveillance systems in LMIC. Did the former question sufficiently capture what is currently happening with FBD-surveillance systems in LMIC?	We have clarified in the manuscript that the focus was FBD surveillance and not FBDs in general. We appreciate the reviewers' comments on the clarity of the workshop questions. Prior to the discussion, workshop participants received a comprehensive introduction to the primary goal of the study: identifying the challenges hindering the implementation and effectiveness of FBD surveillance in African LMICs. This briefing ensured that participants understood the focus on surveillance systems, even though some of the Iceberg questions themselves did not directly reference surveillance. We believe that this context sufficiently oriented participants to respond with an emphasis on FBD surveillance-related challenges.
49	Line 348, food safety is a public good and thus, requires a regulatory framework to ensure compliance. Government and its institutions are critical actors in ensuring a robust food safety and surveillance system. This is clearly missing from your results and discussion.	We agree, and in the new methodology and results section, it will be clear to the reviewers how we have incorporated compliance into the methods.
50	Lines 460-461, sentence is either missing or incomplete.	We have added to the sentence, ensuring that it is not incomplete in the new manuscript.

51	Line 490-497, your results have not demonstrated the lack of these recommendations in the current FBD surveillance systems. The recommendations you make must flow from your results and conclusions.	With the new methodology employed in the manuscript, we have rewritten the results and discussion to ensure that the recommendation flow much more clearly from the results. We hope that the reviewer will agree on this, and we suggest revising the new results section.
52	Line 536, you have not demonstrated in the current study the paradigms that govern public health strategies, which will necessitate a shift. The conclusion must flow from your results.	We agree and we have added to and re-written the results section to more accurately show our results and conclusion.

	Reviewer comment	Response
	Reviewer #1	
1	Thank you for putting so much work into this paper and being so responsive to all reviewer comments. I can see the huge amount of work that went into revising this draft, and I think this is a significant step forward. However, I have three remaining overarching concerns that I go into detail on below: 1. I think the paper needs to be significantly revised structurally to make it more concise and also make sure it flows well. In particular, large portions of the introduction should be combined with the methods section and/or eliminated. 2. This version of the paper has added extensive discussion of dynamic modelling. It's great you've explored this step, but I think this paper is doing too much, and it means that it's still difficult to see how all of your analyses come together to justify your conclusions. My overall recommendation is to remove dynamic modelling from this paper and suggest it as future research, and then work on the dynamic modelling you've done already as a follow up paper, allowing full explanation of the dynamic modelling. 3. The description of how you built your CLDs and what the CLDs told you is much improved in this paper. However, the figures in the Appendix do not seem to follow the conventions of system dynamics / match up well with the text – this needs to be addressed to ensure the paper all aligns.	We sincerely thank the reviewer for their careful reading of the revised manuscript and for acknowledging the extensive work put into addressing the previous comments. We greatly appreciate the constructive feedback and the clear guidance provided regarding the remaining reflections. In response to the first point, we have restructured the manuscript to improve conciseness and flow. Specifically, we have merged sections of the Introduction that were previously theoretical with the Methods section. Regarding the second point, we agree with the reviewer's suggestion that the manuscript was attempting to cover too much by including the dynamic modeling. In line with the advice, we have now removed the dynamic modeling results and discussions from this paper and instead referenced dynamic modeling as a future research direction. Finally, for the third point, we have thoroughly revised the causal loop diagrams (CLDs) to better follow system dynamics conventions. We carefully reviewed the polarity of the relationships, clarified the

		meaning of colors in the figure legends, and ensured that the loop structures consistently match the narrative presented in the main text.
2	Lines 114 - 134– The introduction to systems thinking is quite abrupt, and possibly not justified. However, the description of ‘wicked problems’ is nice – I suggest removing the reference to systems thinking in lines 114 – 118 (possibly just remove these lines completely), and update lines 132 – 134 to explicitly connect the fact that wicked problems can be addressed using systems thinking.	Lines 114-118 has been removed and the connection between wicked problems and systems thinking has been updated.
3	Lines 141 – 145: The definition of system dynamics needs updating. Causal loops are structures that cause system dynamics (patterns of change over time).	The definition in the stated section has been clarified.
4	Lines 149 – 296: I can see all the work you all put in to explain your various methods. Thank you for that. I do think now that the paper has swung too much in the other direction and has begun too long. I have made comments about lines 149 – 296, but after having read the methods, I’ve swung back here, and overarching, my recommendation is to combine lines 149 – 296 with the methods, and streamline.	We have combined lines 149-296 into the Methods section and streamlined the text to ensure that it fits within the flow of the paper.
5	Lines 149 - 152: These lines serve as a transition to wrap up your introduction and start your methods – I suggest doing this clearly and intentionally. Have a clear ending to your intro where you state the aims of your study, and then start a new section as methods. I recognise that you have a methods section later, but this part feels like a methods section more than an introduction.	We agree with the suggestion and have used this section as a natural end of the introduction in the revised manuscript.
6	Lines 155 – 159: Need a brief definition of the iceberg model here.	A brief definition of the Iceberg model has been added.
7	Line 168: The term ‘This organised list” is confusing – needs to be clarified.	“Organized list” has been exchanged with “elements identified in the workshop”.
8	Line 172: I’m not familiar with ‘analysis of exhaustiveness’ – but this doesn’t seem like the right description of what you did. Do you have a citation for the validation step that you did? If	The text has been revised to better explain what we did. After developing the causal loop diagrams

	not, that's okay, but then in that case, in lines 174-175, I suggest being clearer about what you 'cross-checked' against.	(CLDs), we systematically reviewed each loop and variable to ensure that all relevant factors, interactions, and outcomes were considered, and cross-checked them against known components of FBD surveillance systems.
9	Line 176: Is 'Archetype detection' an official process? If so, it needs a citation. Either way, a clearer description of what this step involved is needed.	Archetype identification is an established process in systems thinking, as described by Maani and Cavana (2000) and Senge (2010). In the revised manuscript, we have clarified that we followed the methodology outlined by Senge (2010), which involves assessing the behavior of causal loops to identify recurring system structures (archetypes) and understand the potential consequences these patterns have within the system.
10	Line 181: Did you have a better understanding of the system dynamics or the system structure or both?	The sentence has been changed to clarify the meaning.
11	Lines 184 – 186: dynamic modelling is a complex process – more detail is needed in how you actually carried that out.	As the dynamic modelling has been omitted the revised version, this comment has not been implemented.
12	Lines 187 – 194: This process needs further description and possibly citations.	The scenario planning and implementation step has been clarified in the revised manuscript to further emphasize what was done in the analysis.
13	Lines 207 – 208: Is the term 'structural' needed there given the rest of the list?	We have removed the term "structural".
14	Line 209: I suggest replacing 'the mental structures that people hold' with 'People's deeply held beliefs'	The sentence has been revised.
15	Lines 204 – 296: Thank you for the comprehensive description of all the different	Following the reviewer's suggestion, we have

	theories / tools you used. However, I note that this manuscript has become quite long (57 pages) and would suggest that it may be beneficial to significantly reduce the length of these sections to instead preference more details about what you actually did in the study (in the methods section).	merged the descriptions of the different steps and tools used in the systems thinking approach into the Methods section.
16	Lines 335 – 342: Acknowledging the paper is already quite long, I think you need a (short) supplement section outlining your search strategy for the scoping review + inclusion / exclusion criteria. Updating having read further: I think you have done this in lines 511 – 518, which I recommend moving to the methods section. I recommend having the first two columns (Concept & Search string) in the methods and the second 2 columns (Number of hits & relevant publications) in the results. Alternatively, you could leave all of Table 2 in the results, and just briefly note in the methods that some of these details are in Table 2 in the results.	We have moved the description of the scoping review including the search strategy and inclusion/exclusion criteria into the Methods section. We have decided to keep the full table in the results section and instead referred to the table in the Methods.
17	Lines 349 – 350: The use of centrality in CLDs has been critiqued – you’re already doing so much in this paper, I recommend removing that portion from your methods / analysis. Additionally, you don’t seem to pick this up again in the results, which is even more reason to just remove this line.	Centrality has been omitted in the revised version of the manuscript.
18	Lines 385 – 445: These lines read more as results. Additionally, the equations used here could probably go in an appendix (it’s great you’ve included them!), with just a high level summary of the model + the results of the simulations in the results section of the paper. I would also encourage you to consider removing the dynamic modelling element of this paper, spend more time on finalising your dynamic modelling work, and publish it later as a separate paper. The paper is doing too much (although it’s all great work!) and the dynamic modelling part deserves its own breathing space and room to fully explain.	After the suggestion of the reviewer, we have removed the dynamic modelling section of the paper.
20	Lines 462 – 467: needs citations and/or more details	We have now added a more detailed description of the methodology, both in the overall approach and in the section describing how

		the scenario planning and implementation was carried out. We have explained the validation steps as well.
21	Lines 468 – 471: great that you did this but need more detail – was this in a focus group, interview, workshop, discussion? How many experts?	The scenarios were presented to the same group of participants who attended the original workshop. This has been clarified in the revised manuscript.
22	Lines 525 – 573: These are really nice explanations of your loops, thank you. However, in the supplement / overall map, there are some issues that need to be addressed (see comments on supplement below).	CLDs have been modified in the supplementary material.
23	Table 3: I'd label the first column as 'loops,' not CLDs.	Changed in revised manuscript.
24	Line 655 – 695: Having now read the results here, I feel even more strongly that the dynamic modelling portion should be removed and saved for a future paper. That will give the remaining elements the breathing room and clarity, and also give you the opportunity to report on the dynamic modelling in more detail in a separate publication.	The dynamic modelling has been removed from the paper.
25	Lines 696 – 750: Building on my points on dynamic modelling, if you decide to remove that, I think you could clarify and reduce this section. I'm not clear on if this was further run through the simulation model or whether the comments / table were developed in a different way. If these are further simulation model results, they would require even more explanation, adding even more length to the paper. My advice therefore would be to write this section just based on what arose from the stakeholder engagement / literature review / CLD / leverage point analysis, and then in your discussion, point to dynamic modelling as the next step to develop this further.	The dynamic modelling was used a supplement, and the scenario planning came from the leverage point analyses and the CLDs. The text has been rewritten to clarify this.
26	Similarly to above, while your discussion makes reference to dynamic modelling, it appears the bulk of your discussion comes from all the other steps (besides dynamic modelling) further justifying removing dynamic modelling from this paper and having it as a future paper.	The discussion has been revised to omit the part of dynamic modelling.

27	(Regarding Supplementary Figure S4 – S11): I have a significant amount of experience with building causal loop diagrams, and I found the loops in these figures very hard to interpret. I couldn't figure out which polarity (positive or negative) the red and blue coincided with, and the 'double negatives' that arose from variables like "Inaccurate and incomplete diagnosis"; "Ill-equipped diagnostic facilities"; "Lack of education"; "Underreporting of disease"; "Mistrust in health interventions" compounded this confusion. I found the descriptions of the loops in the main text helpful and meaningful. Reviewing the diagrams, I'm not certain that they align with the conventions of system dynamics and reinforcing and balancing loops, creating uncertainty and confusion about your analysis, including into the archetype and dynamic modelling phase. I think it would be great to redo these CLDs so that they closely follow the conventions of SD and match the text. This effort would also be more important, in my opinion, than expanding the dynamic modelling section, and is further justification to hold over dynamic modelling for a future paper.	We sincerely thank the reviewer for their thoughtful and detailed feedback regarding the causal loop diagrams (CLDs) in Supplementary Figures S4–S11. We appreciate your expertise in system dynamics and agree that the initial presentation of the loops could cause confusion regarding polarity interpretation and consistency with system dynamics conventions. In response to the comment, we have carefully reviewed and substantially revised the CLDs to improve clarity and alignment with established modeling practices. Specifically, we reassessed each causal relationship to ensure the correct assignment of positive and negative polarities and confirmed that the overall structure of each loop matches the behavior described in the manuscript. While we have revised the structure and directionality of the loops to address these concerns, we chose to retain the original wording of the variables to preserve consistency with the workshop outcomes and avoid confusing readers by introducing different terminology than what was used during data collection. We have clarified the color coding in the figure legends so that it is immediately apparent that blue arrows indicate positive relationships and
----	--	---

		red arrows indicate negative relationships
	Reviewer #2	
28	The authors have done a good job revising the manuscript. They have addressed many of the comments, and the current version is a significant improvement on the earlier version. However, further revision is needed to improve the manuscript.	We thank the reviewer for the comment and we have implemented the corrections suggested when appropriate.
29	Unfortunately, the new title of the manuscript does not reflect its content. "...combat inequality..." inequality only appears in the title. The authors do not mention or discuss inequality anywhere in the manuscript except in the title. Based on the content of the manuscript, I suggest a title along the lines of [The Effectiveness of Africa's Foodborne Disease Surveillance Systems: A Systems Thinking and Modelling Approach]	We have changed the title to the suggested format from the reviewer, but omitted the part about the modelling.
30	Line 81, "...tends" should be [trends]	Changed in revised manuscript.
31	Line 90, "locally consumed food", what does it mean? It is confusing. It does imply that then there is internationally consumed food. Locally produced food (line 94) is understandable but not locally consumed food.	The sentence has been reformatted to more accurately reflect the purpose of the sentence.
32	Line 103, "...ignorance of food-safety practices," This is an inaccurate statement. There is enough literature on food safety showing that African consumers are generally knowledgeable about food safety. I suggest you either delete or consider replacing it with [low compliance with food-safety practices]	Sentence has been changed to suggestion by reviewer.
33	Line 105, "antimicrobial resistance", I suggest introducing the abbreviation [AMR] before using just AMR in line 107.	AMR has been correctly abbreviated and used in the correct format in the revised section.
34	Lines 152-296, all this section should be moved under the METHODS sections. Refer to the journal's manuscript style and formatting guide	The section has been moved to the methods section in the revised manuscript.
35	General comment: I don't see the research question clearly stated in the introduction. Line 149-152, "...systematically explore and address the challenges hindering the implementation and effectiveness of FBD surveillance systems in African LMICs". If this is the research question/statement, I suggest you rephrase it	The research question has been rephrased to emphasize the goal of the study and to be more clear about the purpose of the study.

	appropriately and clearly. It is currently not clear enough.	
36	I suggest you move Lines 152-296 under methods. If you think that will make it long, then move some sections to supplementary materials.	The sections have been implemented in the methods section, shortening the paper significantly.
37	I don't see where the discussion ends and the conclusion starts.	We have added a conclusions section to the revised manuscript.
38	Where does the conclusion start? There is no conclusion heading.	We have added a conclusions section to the revised manuscript.
39	Supplementary Table S1, I will not include the names of the workshop participants for confidentiality reasons and pseudonymization.	Names of workshop participants have been removed as suggested.

Manuscript ID: COMMSMED-24-0676A

	Reviewer comment	Response
	Reviewer #3	
1	Line 23; African LMICs? Delete LMICs. Except for Seychelles, no African country is a high-income country. They are either low or middle income countries. Therefore, just write “while previous efforts in Africa have...”	Changed in revised manuscript.
2	Line 31; African LMICs again. Correct accordingly throughout the manuscript.	Changed in revised manuscript.
3	Line 101-105, sentence is too long (52 words). Rephrase it.	Changed in revised manuscript.
4	Line 105-108, sentence is too long (41 words). Rephrase it.	Changed in revised manuscript.
5	General comment: My concern about the research question has not been addressed. The aim of the study is not the same as the research question. The research question must be concise. Example, how do structural, behavioural and contextual barriers limit an effective FBD surveillance system in Africa? A question framed this way allows you to identify the barriers and analyse their pathways to influence/limit effective FBD surveillance systems. That’s what I hope to see from a research question. So, once again, I suggest you frame your research question to reflect your work.	The aim has been rephrased to include a research question, as suggested by the reviewer (line 162-166).
6	Line 257, ...structural, behavioral, or contextual challenges, provide an explanation or a definition of these classifications. Providing very brief definitions will assist the reader know the boundaries within which you are using these words.	A brief explanation of these classifications has been provided (line 304-308).
7	Line 325-326, additional literature...outside scoping review, why will this relevant literature be outside the scoping review? If it is relevant, it should have popped up in the scoping review?	While the scoping review followed a systematic approach with defined search terms and databases, it did not capture all relevant literature due to search string specificity, so the additional literature references was identified through citation tracking and expert recommendations after the initial review was completed. These sources were included because they

		aligned closely with the themes identified and further substantiated the findings.
8	Line 329-331, I'm confused here. What is the difference between literature from the scoping review and complementary literature? If you are finding relevant literature (complementary literature) outside your scoping review, will you not be suffering from missing data bias? I may not be understanding what you mean by complementary literature. Your restriction to only PubMed may account for why you have so many complementary literatures. Be mindful of the effect on your results if you were to include other databases.	The intention of combining the workshop findings with a scoping review and additional complementary literature was to ensure as much coverage as possible. The scoping review was conducted using structured and reproducible search strategies, but when recognizing the complexity of foodborne disease surveillance and the possibility that relevant articles might not be found by the search strings alone, we also included complementary literature to support the findings from the workshop. There is always a risk of missing data in any review process, but this approach was specifically chosen to minimize that risk by going beyond a single source, like the workshop or the scoping review. In that sense, the complementary literature was a deliberate way to be more exhaustive and inclusive in supporting the workshop-derived outcomes.
9	Line 365, "...internal, external, or outside elements..." stick to one word, either use external or outside. Line 271-272, be consistent with the use of inside or outside OR internal or external.	The terms external and outside are intentionally used to refer to different concepts in systems thinking. External elements describe influences that are part of the system's broader environment and interact with it directly, while outside elements refer to factors

		entirely beyond the system's defined boundaries. These are not interchangeable terms, and are used in the correct terminology for systems thinking, so therefore, we have not changed them in the manuscript.
10	Table 2, education and cost-benefit..., "The underlying...lack of food safety awareness and education...". Factually inaccurate statement. You can use inadequate/limited/low food safety awareness... or similar words.	We have changed the wording in the manuscript throughout as well as in the tables.
11	Line 477, "lack of education", similar comment as above. There is enough literature to show households and food handlers in Africa have basic food safety knowledge and awareness, but it may not be enough to sufficiently mitigate foodborne disease risks. To be on the safe side use insufficient/inadequate or similar synonyms.	We have changed the wording in the manuscript throughout as well as in the tables.
12	Make the needed changes in other parts of the manuscript where you argue "lack of education"	We have changed the wording in the manuscript throughout as well as in the tables.
13	Line 520, "...changed", do you mean "change"?	Changed in revised manuscript.
14	Line 565-570, the discussion should be about your results and not rearguing parts of your introduction. This paragraph is not necessarily discussing any part of your results.	Removed in revised manuscript.
15	Line 579, "...handler" should be "handlers"	Changed in revised manuscript.
16	Line 590, "... animals to produce". For better clarity, I would change produce to crops or plants.	Changed in revised manuscript.
17	Line 608, "...FBD surveillance..." I suggest you add "by governments" so, A lack of prioritization of FBD surveillance by governments...	Added in revised manuscript.
18	Line 639-641, I suggest you provide a reference.	Reference added in revised manuscript.
19	Line 653-654, I suggest you delete "these recommendations...analysis"	Removed in revised manuscript.
20	Line 651-668, I suggest recommendations form part of the conclusion.	The recommendations have been incorporated in the conclusions in the revised manuscript (line 859-862).
21	The conclusion is incomplete. The "take away" message or novel finding is missing. I suggest you	The recommendations have been incorporated in the

	incorporate Lines 40-41 and Lines 46-47 in the conclusion and add the recommendations.	conclusions in the revised manuscript (line 865-872).
	Comments on rebuttal to comments of reviewer #1	
22	GENERAL: I read the comments of reviewer #1 and the responses of the authors. Then, I verified the changes made by the authors in the text. Many of the comments raised by the reviewer have been satisfactorily addressed. However, a few comments will require further minor action. Feedback on the rebuttal table: In the future, the authors should indicate where (new line numbers) the changes have been made. COMMENTS: The authors have satisfactorily addressed many of the suggestions, reflecting the improved status of the manuscript. The authors should take further action on the following few issues.	Thank you for the comments. In the revised version, we have ensured to add the line number to the changes for better clarity.
23	Line 159, iceberg definition or description was provided without an appropriate reference. A reference was provided in line 227.	Citations have been added in the revised manuscript.
24	Line 201, "...analysis of parsimony...", provide appropriate reference.	Citations have been added in the revised manuscript.
25	Line 202, "...analysis of exhaustiveness...", provide appropriate reference.	Citations have been added in the revised manuscript.
26	4. Line 235, correct spelling of "... deeply held beliefs..."	Changed in revised manuscript.
27	Table 1 (Search strategy for scoping review...), it is too long (8 pages of a single table?). I suggest it will be appropriate if the whole table is moved to supplementary material and you can keep the first and last columns in the main text.	Table 1 has been moved to supplementary material as suggested by the reviewer and renamed to Supplementary Table 3.
28	Line 311-313, the revised sentence is still not clear. Were the scenarios presented DURING the workshop or were they sent to the participants for comments AFTER the workshop?	The revised sentence has been cleared to more accurately reflect what we did for the participants.
29	Supplementary Figures S4–S11 satisfactorily addressed.	Due to the incomplete sentence, we assume that the reviewer confirms that the supplementary figures have been addressed satisfactory.